# Strong Increase of Thawing of Subsea Permafrost in the 22nd Century Caused by Anthropogenic Climate Change

Stiig Wilkenskjeld[1], Frederieke Miesner[2], Paul P. Overduin[2], Matteo Puglini[1,3,*], and Victor Brovkin[1,4]

[1]Max Planck Institute for Meteorology, Hamburg, Germany.
[2]Alfred Wegener Institute Helmholz Center for Polar and Marine Research, Potsdam, Germany.
[3]Université Libre Bruxelles, Bruxelles, Belgium.
[4]CEN, University of Hamburg, Hamburg, Germany.
[*]No longer active in science.

**Correspondence:** Corresponding author: S. Wilkenskjeld, Land in the Earth System, Max Planck Institute for Meteorology, Bundesstraße 53, D-20146 Hamburg, Germany
(stiig.wilkenskjeld@mpimet.mpg.de)

**Abstract.**

Most Earth System Models (ESMs) neglect climate feedbacks arising from carbon release from thawing permafrost, especially from thawing of subsea permafrost (SSPF). To assess the fate of SSPF in the next 1000 years, we implemented SSPF into JSBACH, the land component of the Max Planck Institute Earth System Model (MPI-ESM). This is the first implementation of SSPF processes in an ESM-component. We investigate three extended scenarios from the $6^{th}$ phase of the Coupled Model Intercomparison Project (CMIP6). In the $21^{st}$ century only small differences are found among the scenarios, but in the upper-end emission scenario SSP5-8.5, especially in the $22^{nd}$ century SSPF ice melting is more than 15 times faster than in the preindustrial period. In this scenario about 35% of total SSPF volume and 34% of SSPF area is lost by year 3000 due to climatic changes. In the more moderate scenarios, the melting rate maximally exceeds that of preindustrial times by a factor of 4 and the climate change induced SSPF loss (volume and area) by year 3000 does not exceed 14%. Our results suggest that the rate of melting of SSPF ice is related to the length of the local open water season, and thus that the easily observable sea ice concentration may be used as a proxy for the change of SSPF.

## 1 Introduction

More than $1300 \, Pg$ carbon are stored in the permafrost soils (perennially frozen soils) of the Arctic (Hugelius et al., 2014). During the present interglacial period, the Holocene, microbiological activity in partial thawed soils degraded a fraction of the stored organic carbon and released it to the atmosphere. Enhanced warming during the Anthropocene has in recent years accelerated this carbon release (Schuur et al., 2015). Most studies have been focused on the thawing of the terrestrial permafrost (Koven et al., 2015; Kleinen and Brovkin, 2018; Turetsky et al., 2020). Since the Last Glacial Maximum (LGM) about $3.5 \cdot 10^6 \, km^2$ (Sayedi et al., 2020) of permafrost soils were submerged by the sea level rising by about $120 \, m$. These submerged permafrost sediments (subsea permafrost, SSPF) now form the major part of the Arctic Shelf. Sayedi et al. (2020) suggest

that about $500\ Pg$ carbon in the form of organic carbon and methane gas may be trapped in the SSPF and become available for microbial decomposition as SSPF thaws. This estimate is however highly uncertain due to the sparsity of measurements (Schuur et al., 2015). Since the benthic temperatures are above the freezing point and their inter-annual variability small, the submergence below sea water causes a slow but continuous thawing of the submerged permafrost sediments from the top. Unless the Earth enters a new glacial state, the SSPF will thus ultimately be thawed away at some time. Anthropogenic climate change may accelerate the thawing and thus the release of carbon.

The quantitative projection of the future climate is often done using Earth System Models (ESMs) which include the carbon cycle. However, among the ESMs participating in the $6^{th}$ phase of the Coupled Model Inter-comparison Project (CMIP6, Eyring et al. (2016)) only few include the dynamics of the terrestrial permafrost (Lawrence et al., 2019) and none of them included subsea permafrost. These models thus lack potentially important feedbacks from the carbon cycle on the global climate. Several modelling studies have aimed at projecting the fate of terrestrial permafrost (Kleinen and Brovkin (2018), McGuire et al. (2018)). A few studies hindcasted subsea permafrost regionally (Shakhova et al. (2009), Nicolsky and Shakhova (2010), Nicolsky et al. (2012), Angelopoulos et al. (2019)) or at the pan-Arctic scale (Overduin et al., 2019). Extrapolating observed temperature trends, Dmitrenko et al. (2011) constructed a future scenario for the SSPF in the Laptev Sea and on the East Siberian Shelf. In this study we present the first steps to include subsea permafrost in the Max Planck Institute Earth System Model (MPI-ESM) to investigate the magnitude of SSPF—climate feedbacks in the future. We explore the possible range of pan-Arctic SSPF thawing by applying climate forcings from several CMIP6 scenarios to the Arctic SSPF areas.

## 2   Methods

The land model JSBACH (Reick et al., 2013; Brovkin et al., 2013) is the land component of the MPI-ESM (Giorgetta et al., 2013). The JSBACH version in the MPI-ESM 1.2 (Mauritsen et al., 2019) includes a multi-layer soil hydrology (Hagemann and Stacke, 2015) and the multi-compartment soil carbon model YASSO (Tuomi et al., 2009; Goll et al., 2015). The version used in this study was extended with a model for freezing soil water (Ekici et al., 2014), the dynamical inundation model TOPMODEL (Beven and Kirkby, 1979; de Vrese et al., 2021) and a methane release model (Riley et al., 2011) as described in Kleinen et al. (2020). de Vrese et al. (2021) furthermore added vertically layered soil carbon to study the organic soil development in permafrost regions. Still modifications were needed to be able to simulate SSPF.

### 2.1   Submerging

To implement SSPF processes in an ESM land model, it is necessary to consider both land and subsea points which experience very different boundary conditions and have different active processes.

For the subsea points MPI-OM (see e.g. Jungclaus et al. (2006, 2013); Mauritsen et al. (2019) for details), the ocean component of MPI-ESM, benthic temperatures were used as an upper boundary condition instead of atmospheric temperatures as are used for terrestrial points. For subsea points, the radiative and hydrological forcings were switched off. The sediment pore space is assumed to be constantly filled with a mixture of water and ice and there is no advection of water.

Plant growth and phenology were disabled for subsea points, leaving these points with a thermally active and carbon-decomposing sediment model.

## 2.2 Geothermal heat flux

In most experiments JSBACH was forced by a geothermal heat flux from beneath. The geothermal heat flux was based on the geologically based data set presented in Davies (2013). This data set was processed for the SuPerMAP model (Overduin et al., 2019) and converted from a flux (units $W\,m^{-2}$) to a geothermal temperature gradient (units $K\,m^{-1}$) assuming the average heat conductivity of the sediment/soil/rock/water/ice mixture of the lowest layers (i.e. low porosity) to be the same as that of the JSBACH bedrock: $2\,W\,K^{-1}\,m^{-1}$. The resulting pan-Arctic geothermal temperature gradient (Fig. S1) ranges from $19\,mK\,m^{-1}$ to $115\,mK\,m^{-1}$ (average $35\,mK\,m^{-1}$). A consequence of this implementation is that the temperature of the lowest model layer, layer $n$, cannot change arbitrarily. For each model point $T_n = T_{n-1} + \frac{d_{n-1}+d_n}{2}GG$, where $T_i$ and $d_i$ are the temperature and thickness of layer $i$ respectively, $n$ the number of vertical layers and $GG$ is the geothermal temperature gradient. This implementation was for a single point version of JSBACH presented in Puglini et al. (2020).

## 2.3 Salinity

SuPerMAP incorporates salt into the newly deposited sediments during periods when the considered model points are inundated into the ocean, i.e. during the interglacials (Overduin et al., 2019). Since the model points have individual inundation histories, the sediments thus have horizontally and vertically variable salinities. The salt distribution from SuPerMAP was preserved in JSBACH in the form of a local freezing point depression: $T_{melt} = -0.054\,K\,kg\,g^{-1}\,S$, where $S$ is the salinity in units of $g\,kg^{-1}$. Diffusion of salt as modelled by e.g. Angelopoulos et al. (2019); Malakhova and Eliseev (2020) is a very slow process (Harrison and Osterkamp (1978) reports a diffusion coefficient for salinity about 4 orders of magnitude lower than for temperature). Due to the comparatively short time scale covered in the present study diffusion of salt was not implemented.

Despite the slowness of the salinity diffusion, the upper shallow sediment layers may be intruded on short time scales. This causes the model artifact, that the upper layer porewater may seasonally freeze — especially during the winter season — if it is submerged under (saline) liquid sub-zero-temperature seawater. In reality this porewater would, though diffusive exchange with the overlying seawater, be sufficiently saline to prevent freezing. To prevent this buffering of energy in a non-physical re-freezing of the pore water in the upper layers, the active freezing of porewater was disabled for subsea points. Additional experiments were performed to test the consequences of this hypothesis.

Implementing the freezing point depression due to salt allows two different definitions of SSPF — according to temperature or according to ice content. Here, we refer to SSPF only when the sediments are frozen, i.e. it is not sufficient to have sub-zero temperatures.

## 2.4 Model setup and experiments

JSBACH was run on the T63 horizontal grid with a resolution of $\approx 1.9 \times 1.9$ degrees or $\approx 50 \times 200\,km$ at the Arctic latitudes around $75°N$ where most of the SSPF is found. In this setup, 407 points are located on the modern Arctic shelf at a distance of $\leq 20\,km$ from a point in the SuPerMAP model (Overduin et al., 2019). In the SuPerMAP setup, points with water with depths $< 150\,m$ were selected since these potentially hold SSPF. In this JSBACH setup a subsea point on average represents an area of about $11,400\,km^2$. Vertically, 22 layers were used to represent the upper $1000\,m$ of sediment with layer thicknesses between $6.5\,cm$ and $300\,m$. Generally the layer thickness increases exponentially with increasing depth, but additional layers were added to the depth range $0.5 - 3\,m$. This vertical setup is a downward extension of the 18 soil layer model used by de Vrese et al. (2021).

Sediment porosity and water depth were adopted from the SuPerMAP points used. The porosity decreases exponentially with increasing depth from $\approx 0.4$ in the upper layer to $\approx 0.1$ in the lowest and is the same for all geographical points.

The thermal sediment properties were assumed to be those of the standard JSBACH "bedrock" for which the effective diffusion is calculated as a mixture of rock and water weighted with the porosity (Ekici et al., 2014; de Vrese et al., 2021).

Experiments were conducted following each of the three CMIP6 SSP scenarios SSP1-2.6, SSP2-4.5 and SSP5-8.5 to evaluate the fate of SSPF ice in scenarios with low, intermediate and upper-end future global warming, respectively.

Since SSFP is never in equilibrium with the boundary conditions, two additional control experiments were conducted, referring to a preindustrial and a present day climate respectively. These two experiments are used to assess SSPF thawing in a hypothetical world without anthropogenic climate change and were cyclically forced with 24 years of forcing data from 1850-1873 (preindustrial) and 1986-2009 (present day) respectively. Further experiments were performed to assess the consequences of some of the model assumptions.

Most of the performed experiments (Table 1) cover at least the period 1850 to 3000. The exceptions are the present-day control simulation which branches off from SSP1-2.6 in 2010 and thus covers only the years 2010 to 3000, and some of the sensitivity runs. The three main experiments, *pmt_ssp126*, *pmt_ssp245* and *pmt_ssp585* are identical for the period 1850 to 2009.

## 2.5 Initial conditions

The model was initialized with sediment ice content, temperature and salinity from SuPerMAP (Overduin et al., 2019). For each point the data from the SuPerMAP point nearest to the center of the respective JSBACH cell was used.

We took advantage of the elaborated spin-up protocol applied to the SuPerMAP model (Overduin et al., 2019) which uses $450\,kyr$ average temperatures from CLIMBER-2 and geothermal heat flux to calculate steady state profiles of temperature and SSPF ice for each point. This is longer than the $\approx 100\,kyr$ estimated by Malakhova and Eliseev (2017) to be necessary for a proper equilibration of the deep sediments. This steady state solution was used as initialization for $50\,kyr$ long runs using idealized benthic temperatures (ranging from $-1.8$ to $0\,{}^{o}C$, dependent on water depth) and the point individual transient submergence history. To minimize artificial initial effects arising due to the transition from SuPerMAP (with its idealized upper

boundary condition) to JSBACH (using the MPI-ESM data as forcing) the last $23,5\,kyr$ (since the LGM) of the SuPerMAP run were rerun using a linear interpolation between the idealized benthic temperatures and the temporal average of the MPI-ESM CMIP6 preindustrial benthic temperatures, defined as the period $1850 - 1873$. The interpolation was ended $1\,ka\,BP$, so that, for the last 1000 years, SuPerMAP was forced solely with the preindustrial temperatures. Even though this approach largely eliminates transient effects, additionally $\approx 500\,km^3$ SSPF ice melts until year 2000 compared to what could be expected from the control experiment. More than 80% of the additional melting takes place until year 1900 which is also about the time where the melting in the climate driven scenarios start exceeding that of the control simulation. Since this effect is small and our focus is on the future scenarios, we regard these minor artifacts as acceptable and without influence on our main results.

The SuPerMAP output at $2\,m$ vertical resolution, was assumed to be valid at the layer midpoint and interpolated linearly to the JSBACH layers. For JSBACH layers above $1\,m$, the values for the upper SuPerMAP layer were assumed representative. Since the seasonal temperature cycle penetrates further down (Fig. S13), the results are not sensitive to this initial assumption.

The ice saturations given by SuPerMAP were converted to "ice column lengths" for each layer $i$ according to $ice_i = \frac{sat_i}{por_i d_i}$ where $ice_i$ is the ice column length, $sat_i$ is the ice saturation, $por_i$ the porosity and $d_i$ the layer thickness. The initial ice column length accumulated over all layers (Fig. 1, left panel) ranges from 0 to over $180\,m$. The initial area of the grid cells where SSPF ice is found at any depth in the sediments (initial SSPF area) is $\approx 2.89 \cdot 10^6\,km^2$ (242 of the 407 points) where $\approx 0.77 \cdot 10^6\,km^2$ is less than $10\,m$ thick. The total initial ice volume is $\approx 153 \cdot 10^3\,km^3$. Vertically, the ice saturation (Fig. 1, right panel), is highest at $\approx 6\,m$ depth with a secondary peak at $\approx 75\,m$. The SSPF ice saturation is closely anticorrelated with the embedded salt, consistent with the freezing point depression and the inundation history of the last two glacial cycles.

## 2.6   Boundary conditions

The subsea points of JSBACH were forced with monthly mean benthic temperatures as upper boundary conditions. These were adopted from the MPI-ESM runs presented in detail in Kleinen et al. (2021) extending the CMIP6 scenarios beyond year 3000. Until year 2500 these runs were forced by the $CO_2$ concentration scenarios presented in Meinshausen et al. (2020). After year 2500 MPI-ESM was forced by $CO_2$ concentrations from the intermediate complexity climate model CLIMBER-2 (Brovkin et al., 2012; Kleinen et al., 2016; Petoukhov et al., 2000). The temperature of the lowest of the 40 fixed depth MPI-OM ocean model layers above the ocean floor was assumed to be the benthic temperature. This was interpolated from the GR30 grid (resolution varying within the SSPF area from $\approx 40\,km$ near Greenland to $\approx 200\,km$ on the East Siberian Arctic Shelf) onto the T63 grid using *CDO remapcon* (Schulzweida, 2019). Despite the rather coarse oceanic resolution in the Russian Arctic, MPI-OM reproduces the trend in the coastal Laptev Sea benthic temperature reported by Dmitrenko et al. (2011) remarkably well ($2.1\,K$ reported vs. $2.15\,K$ in MPI-OM during the period 1985 to 2009, not shown). Our data however suggest that this strong trend is confined to the Laptev Sea area and temporarily limited to a period not much longer than the reporting period of Dmitrenko et al. (2011). The forcing of the different scenarios diverge from 2010 onward (not from 2016 as described in the CMIP6 protocol (Eyring et al., 2016)) for technical reasons (see Kleinen et al. (2021)).

Three scenarios were used: SSP1-2.6, SSP2-4.5 and SSP5-8.5. In all scenarios the benthic temperature rises from around year 2000 and stabilizes at a new essentially constant level (Fig. 2). In SSP1-2.6 the rise only lasts a few decades and is in total

150 $< 1\,K$, in SSP2-4.5 the temperature in total rises $\approx 2\,K$, mainly before year 2150. For these scenarios, the largest temperature increases are found in the Barents Sea and Baffin Bay (Fig. S5), closely relating to the atmospheric temperature rise (Fig. S4), where not much SSPF ice is found (Fig. 1, left). The SSP5-8.5 benthic temperature in total rises almost $10\,K$ with the strongest trend in the $22^{nd}$ century (Fig. 2). For the individual model points the increase is between 7 and $16\,K$, highest in Baffin Bay and lowest in the Laptev Sea (Fig. S5, right). In all scenarios the benthic temperatures increase less than average for points

which contain SSPF and the highest temperature rise is found in coastal waters.

## 3 Results

The amount of SSPF ice decreases smoothly over the entire experiment period (Fig. 3) in all scenario and control experiments, however at very different rates. In the experiment forced with a preindustrial forcing (*pmt_pre*) ice melts at an essentially constant rate of $\approx 7.5\,km^3\,a^{-1}$ over the entire period. Virtually same behaviour at rates of 9.1 and $11.7\,km^3\,a^{-1}$ is seen in the

160 present day (*pmt_curr*) and SSP1-2.6 (*pmt_ssp126*) experiments, respectively. The experiments forced with the two SSPs with stronger warming, *pmt_ssp245* and *pmt_ssp585*, result in an accelerating ice-melting from the present day to 50–100 years in the future, and thereafter slowly decreasing melting rates. The average melting rate for the *pmt_ssp245* is $17.7\,km^3\,a^{-1}$ and thus not too much higher than for the stabilization scenarios, *pmt_curr* and *pmt_ssp126*. However, the *pmt_ssp585* has an average melting rate of $54.6\,km^3\,a^{-1}$ and thus on average more than 3 times as much ice is lost, even compared to the

165 intermediate forcing scenario *pmt_ssp245*. The difference between the scenarios is minor in the $21^{st}$ century, but the melting increases dramatically the $22^{nd}$ in the *pmt_ssp245* simulation and much more so in the *pmt_ssp585* simulation. The ratio between the SSPF ice meltings in a scenario run and in the control run (*pmt_pre*) emphasizes the influence of climate change on the SSPF. During the $22nd$ century this ratio averages to 15.8 for *pmt_ssp585* (Fig. 4), where on average $116\,km^3\,a^{-1}$ of ice is lost. *pmt_ssp245* melts 3–5 times more than the control. The ratio drops again to about 7 and 2 for the two scenarios

respectively and even further in the far future beyond 2500. In *pmt_ssp126* and *pmt_ssp245* the main melting takes place on the shelf of the Kara sea (Fig. 5) whereas in *pmt_ssp585* the main ice loss is on the shelves of the Laptev and East Siberian Seas. The geographic patterns of the melting are essentially constant over time (not shown). In agreement with earlier findings (Overduin et al., 2012, 2019; Angelopoulos et al., 2019), the largest melting is found near the coast, where the water is shallowest and the submerging took place most recently. Thawing in these regions was initiated comparatively recently, and thus the ice-bonded

permafrost table (IBPT) is still close to the sediment-water interface.

By the year 3000 the initial area of SSPF ($2.89 \cdot 10^6\,km^2$) is reduced to $2.63 \cdot 10^6\,km^2$, $2.55 \cdot 10^6\,km^2$, $2.37 \cdot 10^6\,km^2$, $2.22 \cdot 10^6\,km^2$ and $1.65 \cdot 10^6\,km^2$ for the two control runs and the three SSPs, respectively. The disproportional large loss of SSPF area between 1850 and 2100 (Table 2) compared to the volume loss is explained by the rather large area of thin SSPF ice ($0.77 \cdot 10^6\,km^2$ has $< 10\,m$ of ice). In general, the relative loss of SSPF area is larger than that of SSPF volume, most

180 pronounced for the intermediate scenario experiments *pmt_ssp126* and *pmt_ssp245*. The two control experiments (*pmt_pre* and *pmt_curr*) apply a cyclic forcing which means that there are no long term geographical shifts in the benthic temperatures. This leads to essentially fixed area to volume melting ratio.

Initially the highest SSPF ice saturation ($\approx 86\%$, Fig. 1b) is found in a layer around $6\,m$ depth. In all main experiments the ice saturation of this layer is reduced from about 2050 onward (Fig. 6), but to a very different degree. In *pmt_ssp1-2.6* and *pmt_ssp2-4.5* it is reduced to a saturation of $\approx 55\%$ and $\approx 30\%$ in year 3000 respectively, whereas in *pmt_ssp5-8.5* the ice at this depth is completely melted shortly after year 2100. This layer effectively insulates the deeper ice and thus prevents ice melting in the deeper layers. In *pmt_ssp585* the IBPT quickly deepens after year 2100 and reaches a depth of $\approx 100\,m$ before year 2500. The melting rate of SSPF ice however decreases during this period due to the lower concentrations of SSPF ice between 6 and $100\,m$ depth.

SSPF ice is melted from below by geothermal heat in addition to the climatically driven melting from the above. All ice in the lowest layer (layer 22, center at $850\,m$ depth, initially $0.69\,\%$ saturated) is gone by year 3000, and in layer 21 (center depth $550\,m$) the ice saturation decreases from an initial $14.5\,\%$ to $12.2\,\%$ in year 3000. Effects of geothermal heat flux are also seen in the above layer(s), but in these layers the melting is caused by a combination of heat from below (geothermal) and above (climate).

The development of melting and temperature for a selected section across the East Siberian Shelf in *pmt_ssp585* is illustrated in detail in Fig. 7, where 7 time slices of (upper panel) SSPF ice saturation and sea ice concentration (SIC) and sediment and (lower panel) air temperature changes are shown. Before year 2200 salinity-induced taliks (unfrozen lenses) are found in the layers centered at $38\,m$ and $150\,m$. By year 2100 only a little ice at $2-3\,m$ depth is melted away and the temperature at the same depths rises by $\approx 2\,K$ preferentially towards the coast. In the $22^{nd}$ century a dramatic rise in temperature of $> 8\,K$ in the sediments above $\approx 5\,m$ is associated with melting of all ice above $10\,m$. As long as sea ice is present (until about 2100) the temperature rise in the upper sediments is decoupled from the atmospheric temperature rise. After the disappearance of the sea ice, the temperature rise of upper sediments and atmosphere are closely linked together.

The insulating, and thus SSPF preserving, effect of sea ice can also be observed on the pan-Arctic scale (see Fig. S10 and compare Figs. 2, S2 and S3). In the $19^{th}$ and early $20^{th}$ century the annual mean SIC of the SSPF area is 74% (68% of the modelled area, Fig. S2), rapidly dropping in the first half of the $21^{st}$ century. In SSP1-2.6 this is followed by a slight rise until about 2200 and a stabilization at $> 50\%$. SSP2-4.5 also stabilizes though below $40\%$ after a drop at a lower rate until about 2150. In SSP5.8-5 the sea ice continues to shrink at a fast rate and is entirely gone at the end of the $21^{st}$ century except from seasonal ice in the coastal waters of the Laptev sea. Thus, from this point in time also at pan-Arctic scale, the benthic temperature (Fig. 2) start to adapt to the higher atmospheric temperatures (Fig. S3).

The melting of SSPF ice from above is determined by the energy input into the sediments. A quite direct measure for this energy input is the difference between the benthic temperature and the temperature of the upper sediment layer ($\Delta T$). $\Delta T$ shows clear patterns when plotted against the local SIC (in time and space), Fig. 8. The local SIC sets a limit for $\Delta T$, which is always essentially $0\,K$ when SIC $> 70\%$. Above this limit, the insulating effect of the sea ice seems to be sufficient to prevent rising temperatures in the water column and thus the benthic water is close to thermodynamical equilibrium with the upper sediment. With lower SIC the maximum "allowed" $\Delta T$ rises approximately linearly, independently of the chosen scenario. The most ($\Delta T$, $SIC$) points with low SIC are found in *pmt_ssp585*, but the relation for the maximum $\Delta T$ seems to

be scenario-independent. This "universal" relation may explain the much larger SSPF melting in *pmt_ssp585* compared to the other scenarios.

## 4 Discussion

### 4.1 Relation between SSPF ice melting and sea ice

The rate of SSPF ice melting from above is determined largely by the benthic temperature (Fig. 8). The benthic temperature in turn is limited by the presence of sea ice which suppresses energy input into the ocean. Thus a direct relation can be found between the open sea season and the melting of SSPF ice (Figs. S9, S10 and S11). A similar link was suggested by Dmitrenko et al. (2011) based on recent observed rise in the benthic temperatures on the East Siberian Shelf. Fig. S11 also illustrates the decreasing melting rates from the end of the $21^{st}$ century due to the complete disappearance of SSPF ice near the benthic surface. In our model the link between absence of sea ice and melting of SSPF ice may be exaggerated by the coarse resolution of MPI-OM implying a zero-curtain effect on the sea ice—oceanic temperature relation on a relatively large scale. As long as a model grid cell has $SIC > 0$ all energy which goes into the ocean in this grid cell is used to melt the remaining sea ice, preventing a rise in the water temperature. In reality this is very dependent on the heterogeneity of the sea ice. If the remaining sea ice is collected in a corner of a the cell the water may be heated to above freezing point temperatures in the rest of the grid cell. Also the coarse MPI-OM resolution may underestimate the oceanic advection on the Arctic shelf, exaggerating the influence of sea ice on the SSPF on the local scale. Despite these model artifacts, the basic physics of the causal relation chain from disappearance of sea ice to melting SSPF ice seems plausible.

### 4.2 Model limitations and assumptions

We are ignoring the thermal coupling between sea surface and the sediment at the sea bottom caused by bottom-fast ice (Nicolsky et al., 2012; Osterkamp et al., 1989), since such effects are, due to the coarse resolution of the ESM-setup of JSBACH (Sec. 2.4), never relevant for a significant part of a grid cell.

Freezing of the upper part of subsea sediment below shallow waters is in reality possible (Osterkamp et al., 1989), though this is mainly observed were bottom-fast ice is present. Due to the lack of diffusion of salt in the sediments in our model, we made the assumption that freezing of the below sea sediments is not happening, since in reality the salinity of the porewater of the upper sediments would be in equilibrium with the liquid sea water above. The validity of this assumption decreases with increasing depth into the sediments due to the larger time lag of the salinity entrainment caused by the difference between temperature and salinity diffusivities (Harrison and Osterkamp, 1978)). We tested the consequences of this assumption by a series of sensitivity experiments (Table 1) allowing the entire sediment column (*pmt_freeze* and *pmt_freeze126*) or the sediment below $30\ cm$ (from layer 4 downward) to freeze (*pmt_fr3*). These experiments shows that freezing occurs mainly in the upper 2–3 layers (not shown). The water volume frozen (years 1850–3000) is 41, 170 and 12 $km^3$ for the three experiments respectively. This is very small compared to the total volume of SSPF ice ($\approx 90 - 150 \cdot 10^3\ km^3$). Due to the energy buffering effect of the

freezing/melting cycle in the upper layers, a much larger volume of SSPF ice is preserved: $2.5 \cdot 10^3 \ km^3$ less SSPF ice melts by year 2000 (Fig. S7). For the strong forcing experiments (*pmt_freeze* and *pmt_fr3*) the difference to *pmt_ssp585* is decreasing after year 2000 and is almost gone after year 2300. This "catching up" of the melting is facilitated by the rising benthic temperatures which essentially prevents any freezing of sediments in the later part of these experiments. In the low forcing experiment, *pmt_freeze126*, freezing takes place at an essentially constant rate throughout the experiments, preserving the energy buffering in the upper layers. This results in a slight but persistently lower rate of SSPF ice melting than in *pmt_ssp126*. In total, the difference to the main experiments is at most $\approx 2\%$ of the total SSPF ice volume. Based on these sensitivity experiments, we conclude, that the influence of the "no re-freeze" assumption is minor and does not alter our main conclusions. The "all-can-refreeze" and the main experiment bounds the realistic melting from below and above respectively.

In this study, as well as in the study by Overduin et al. (2019), the effects of increasing hydrostatic pressure with increasing depth on the freezing point were ignored. A preliminary estimate suggests that these effects are almost an order of magnitude smaller ($0.35 \ K$ at $500 \ m$, not shown) than the freezing point depression by salinity. Most of the SSPF ice is found above $500 \ m$ so the effects would be even smaller. With this reasoning Overduin et al. (2019) did not include these effects and including them here would thus also increase the inter-model inconsistency. Including a freezing point depression by hydrostatic pressure would likely introduce an enhanced initial melting of SSPF ice which would in parts be compensated by lower melting rates later on since sediment temperatures would be lower due to additional energy needed for the additional initial ice melting.

In our setup, we suppress water advection in the unfrozen parts of the sediments. This is however unlikely to make any difference for our main focus on the thawing from above, since the main effect is a heating from above which would tend to stabilize the unfrozen sediment water column in the sediments. An active hydrology may however enhance the influence of the geothermal heating by increasing the effective upward heat flux.

On the Arctic shelf, a broad range of sedimentation rates have been reported. From previous works, Overduin et al. (2019) report rates from near zero to $700 \, cm \, ka^{-1}$. For their model, they chose $30 \, cm \, ka^{-1}$ for submerged points. Over the experimental time of the present study, this would lead to a sedimentation of $\approx 34.5 \, cm$ of additional sediment or $12 \, cm$ during the first 400 years of the experiments where the most interesting dynamics take place. Compared to a typical depth of ice melting of at least $8 \, m$ (Fig. 6) and the typical depth to which the seasonal temperature changes penetrates (30% of surface amplitude at $\approx 4 \, m$, Fig. S13), the added sediment is thus small and we therefore neglected sedimentation in the present study. The penetration depth is remarkably constant taken the large temperature range of seasonal amplitudes found at different locations (Fig. S12).

The thermal properties of the (dry) sediments were, for consistency with other studies using JSBACH and MPI-ESM, kept equal to the properties of the standard JSBACH "bedrock" (conductivity $2 \ W/(Km)$, capacity $2 \cdot 10^6 \ J/(Km^3)$). These values may not be adequate for subsea sediments, so — inspired by the measurements of Goto et al. (2017) in muddy sediments of Japan — we conducted two additional experiments (*pmt_pre_lc* and *pmt_ssp585_lc*), setting the heat conductivity for (dry) sediments to $0.8 \ W/(Km)$ and their heat capacity to $3.6 \cdot 10^6 \ J/(Km^3)$). Adding additional pore water to these values, the sediment thermal properties in these experiments are quite close to those of pure water. Thus these experiments can be regarded as an extreme case for the lowest possible heat entrainment into the sediments. The results (Figs. S8 and S9, equivalent to Figs.

3 and 4 respectively) reveal that the changed sediment properties lead to a much lower melting of SSPF ice. Until year 2500 only 38% and 57% are melted compared to the respective main experiments (*pmt_pre* and *pmt_ssp585* respectively). Thus indeed less SSPF ice will melt and less carbon released using these parameters. However, the relative ratio between the main and the control experiment (Fig. S9) even increases compared to the experiments using standard sediment thermal properties, emphasizing the influence of climate change. What remains largely unchanged is the time at which the melting starts to accelerate in the high emission scenario. Also in *pmt_ssp585_lc* this is around 2100, leading to extreme melting rates in the $22^{nd}$ century $\approx 2.5$ times those found in the moderate forcing scenario *pmt_ssp245* which used the standard sediment thermal properties.

## 4.3 Area of SSPF

Our present day (2020) SSPF ice area, $\approx 2.74 \cdot 10^6 \ km^2$, is larger than the SSPF area reported by Overduin et al. (2019), $2.48 \cdot 10^6 \ km^2$ which in turn is somewhat larger than the $\approx 2 \cdot 10^6 \ km^2$ presented in Sayedi et al. (2020). Part of this difference may be explained by Overduin et al. (2019) presenting a preindustrial estimate while Sayedi et al. (2020) refers to present-day. In our model $\approx 0.15 \cdot 10^6 \ km^2$ of SSPF ice area disappears during the 170 year period in between. 76% of this retreat is also found in the control experiment *pmt_pre* indicating that about $\approx 0.036 \cdot 10^6 \ km^2$ of SSPF ice melts away due to climatic changes within the historical period. As discussed in Overduin et al. (2019) also the idealized benthic temperatures used for the SuPerMAP runs are likely to be to cold in regions of oceanic inflow in to the Arctic Ocean such as downstream of the Gulf Stream current system and in the vicinity of the Bering Strait. This may result in larger areas with SSPF in both their and our study than observationally based studies. Both Overduin et al. (2019) and Sayedi et al. (2020) use the more common permafrost definition, based on continuous years of sub-zero temperatures. If we apply this method, our SSPF area is even larger: $3.31 \cdot 10^6 \ km^2$ in 1850 ($3.18 \cdot 10^6 \ km^2$ in 2020). That our SSPF area estimates are larger than the previous studies is likely a result of the coarse T63 resolution causing extrapolation of SSPF to locations further off-shore (by up to $\approx 90 \ km$) than included in the model of Overduin et al. (2019). A $90 \ km$ wide stripe extending around the pole at $75°N$ has an area of $0.93 \cdot 10^6 \ km^2$. Within this margin, our areas are in agreement with the other estimates.

## 4.4 Geothermal heat flux and thawing rates (rate of change of IBPT)

The two sensitivity experiments excluding the geothermal heat flux (*pmt_pre_0* and *pmt_ssp585_0*) melt $4.7 \cdot 10^3 \ km^3$ and $4.9 \cdot 10^3 \ km^3$ ($4.0 \ km^3 \ a^{-1}$ and $4.2 \ km^3 \ a^{-1}$), respectively, less than the experiments with geothermal heat flux (Fig. S6). For *ssp_pre* this is more than half of the total melting, indicating that geothermal heat flux is more important than climate forcing for the melting. These two experiments represent the extremes of the forcing and since the geothermal melting is virtually identical, it can be concluded that climate-caused melting from above and geothermally-caused melting from below are largely independent from each other and thus can be treated independently.

The rate of change of the IBPT controls the rate at which carbon become available for degradation. It can be roughly estimated by $\Delta IBPT = (T_{total} - T_{noheat}) / \overline{por}$, where $T_{total}$ and $T_{noheat}$ are the SSPF ice melting rates in the main experiment and the experiment without geothermal heat flux, respectively, and $\overline{por}$ is the average porosity over the layers where SSPF is

thawed. Assuming $\overline{por} = 0.35$ and ignoring the shrinking of SSPF area with time, the average $\Delta IBPT$ for points with SSPF ice is $0.8\,cm\,a^{-1}$, $0.9\,cm\,a^{-1}$, $1.2\,cm\,a^{-1}$, $1.7\,cm\,a^{-1}$ and $5.3\,cm\,a^{-1}$ for $pmt\_pre$, $pmt\_curr$, $pmt\_ssp126$, $pmt\_ssp245$ and $pmt\_ssp585$ respectively for the 1850–3000 period. The $\Delta IBPT$ for the 1850–2009 period is $1.2\,cm\,a^{-1}$ and can be compared to present-day measurements. Overduin et al. (2015) measured thawing rates between 0 and $16\,cm\,a^{-1}$ off the Muostakh

Island and Overduin et al. (2012) $0.5 - 1.4\,cm\,a^{-1}$ off Barrow. Both these study sites were in shallow waters very close to the coast. The model hindcast study of Angelopoulos et al. (2019) presented thawing rates of $1.2 - 1.5\,cm\,a^{-1}$ and thus very similar to the rates found there. In the $22^{nd}$ century peak of $pmt\_ssp585$ the thaw rate estimate is $11.4\,cm\,a^{-1}$.

## 5    Conclusions

Here, for the first time, a land component of an ESM was used to project the development of subsea permafrost (SSPF) ice

until year 3000 by forcing it with extended CMIP6 scenarios spanning the likely range of climate change. Using an ESM component has the advantage over more specific SSPF models that it can be directly coupled to the carbon cycle to study climatic feedbacks. Though the inter-scenario differences are minor in the present century, in the $22^{nd}$ century, the strong forcing scenario SSP5-8.5 strongly diverges by melting all SSPF ice above $\approx 100\,m$ depth, whereas the moderate and low forcing scenarios still leave SSPF ice below $\approx 8\,m$. This highlights the need to go beyond 2100, the usual time scale on which

climate projections are made, to get the response of slowly acting climate components such as SSPF to a changing climate. The large loss of SSPF ice in SSP5-8.5 is closely linked to the disappearance of local sea ice. Specifically, the length of the open water season controls the temperature rise of the ocean water on the Arctic shelf. This temperature rise in turn determines the energy input into the ocean bottom sediments and thus the melting of SSPF ice. Therefore our results suggest, that the length of the local open water season may be used as an easy observable proxy for the rate of melting of SSPF ice.

*Code availability.*  The MPI-ESM licenced model source code including the JSBACH version used for this study is stored in the git system of MPI and will be made available by the first author on request. The running and plotting scripts are available together with the model output.

*Data availability.*  https://cera-www.dkrz.de/WDCC/ui/cerasearch/entry?acronym=DKRZ_LTA_1142_ds00001

*Author contributions.*  S. Wilkenskjeld made the current subsea implementation in JSBACH, decided on experiment setup, performed the JSBACH experiments and subsequent analysis as well as being the main article writer, F. Miesner set up and performed the SuPerMAP runs

to create the initial data for JSBACH, P.P. Overduin supervised the SuPerMAP work and provided valuable discussion input, M. Puglini made the initial subsea implementation (point model version) and the implementation of geothermal heat flux into JSBACH and V. Brovkin provided initial ideas and supervised the work from start to end. All authors contributed to the writing.

*Competing interests.* The authors declare that they have no conflict of interest.

*Acknowledgements.* This study was funded by the EU-Horizon 2020 projects Nunataryuk, Grant-no. 773421 and CRESCENDO, Grant-no.
641816 as well as the German Research Foundation through the CLICCS Clusters of Excellence (DFG EXC 2037). This work used resources
of the Deutsches Klimarechenzentrum (DKRZ) granted by its Scientific Steering Committee (WLA) under project ID bm1142. The used
model code is based on the version described in de Vrese et al. (2021) which was kindly provided by the Philipp de-Vrese and the forcing
data were kindly provided by Thomas Kleinen. Dirk Notz provided insights into the workings of the MPI-OM model. A. Eliseev and an
anonymous reviewer contributed many valuable comments and suggestions leading to a clearer manuscript and higher confidence in the
results.

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

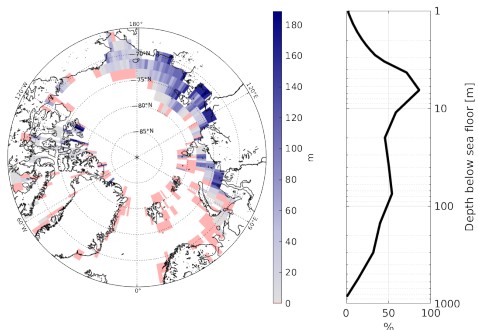

**Figure 1.** *(Left)* Length of initial (1850) SSPF ice column. Redish color mark points entirely without ice in the sediments. *(Right)* Vertical distribution of initial ice concentration in sediment pore space (SSPF ice saturation). The right panel shows the spatial average over the area containing SSPF ice in 1850.

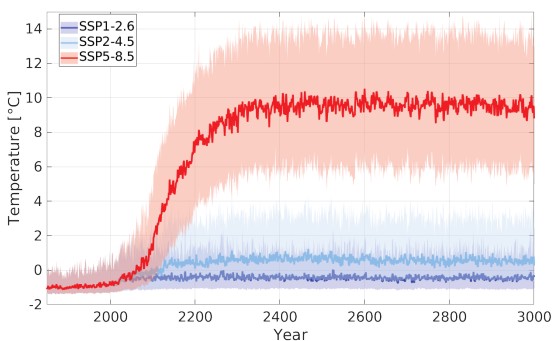

**Figure 2.** Benthic temperature. Lines are yearly averages over the modelled area. Shaded areas show decadal means of the extreme months for each year and are combined RGB-wise.

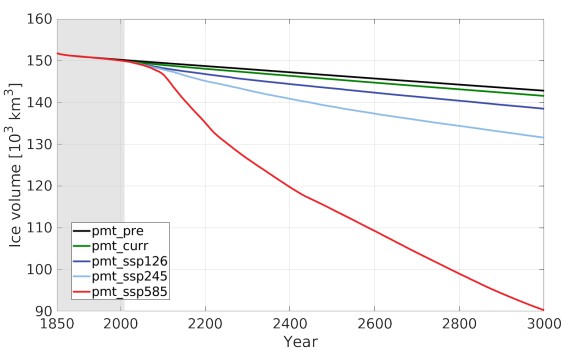

**Figure 3.** Volume of pan-Arctic SSPF ice for the scenario and control experiments. Gray shading marks the historical period.

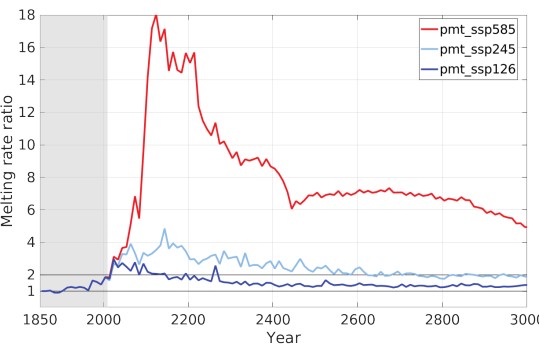

**Figure 4.** Melting rate ratios relative to *pmt_pre* for the main scenarios. 10 year averages. Gray shading marks the historical period.

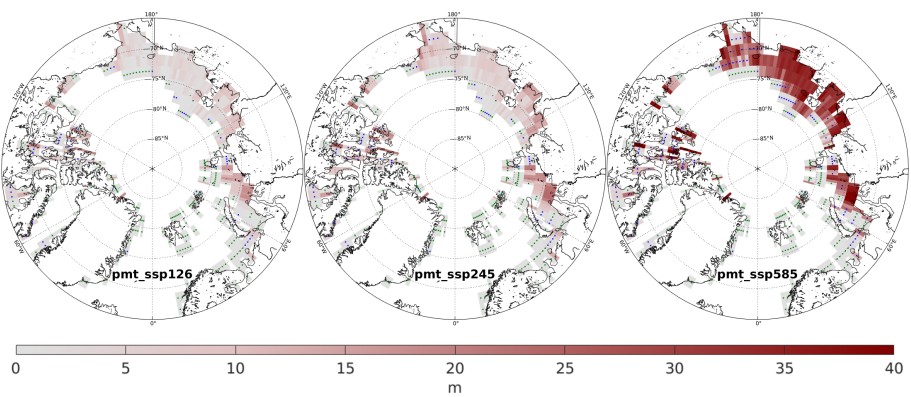

**Figure 5.** SSPF ice column melting for the three main experiments in the period 1850–3000. Green dots mark points entirely without SSPF ice and blue mark points where all SSPF ice melted before year 3000.

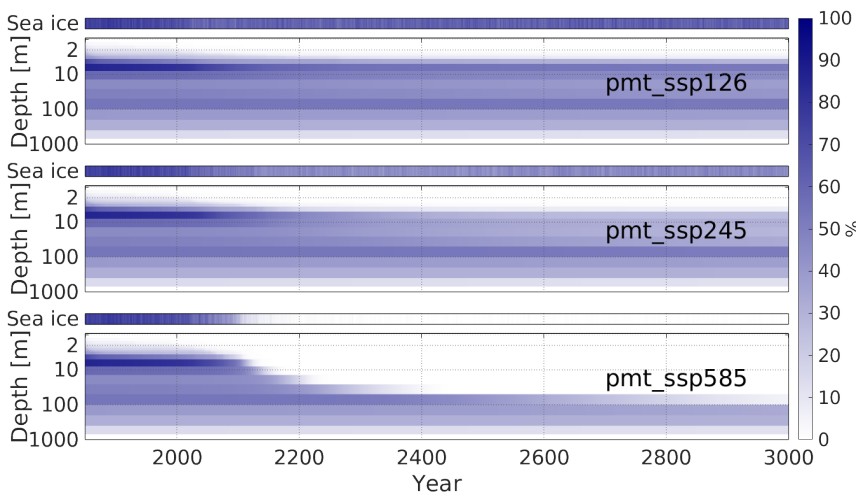

**Figure 6.** Vertical distribution of SSPF ice saturation (main panels) and sea ice concentration (top panels) for the scenario experiments. Yearly averages for the 242 points which contained ice in the initial conditions (1850). "Depth" is measured downward from the sea floor.

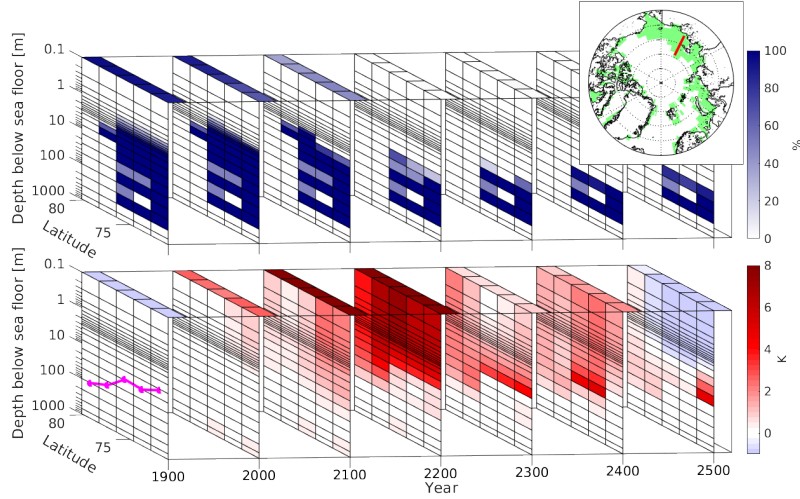

**Figure 7.** SSPF ice saturation (upper panel, *lat/depth*-planes), sea ice concentration (upper panel, *lat/time*-plane), sediment temperature change (lower panel, *lat/depth*-planes) and atmospheric $2\,m$ temperature change (lower panel, *lat/time*-plane) along a section across the East Siberian Shelf ($153^\circ E$, shown on the insert in the upper panel) for different time periods. In the lower panel time slice at 1900 represent the 1850–1900 change, the remaining the changes relative to the slice before. Black lines at constant depth are model layer boundaries. Magenta line shows the water depth along the section. Shown are (differences between) yearly averages.

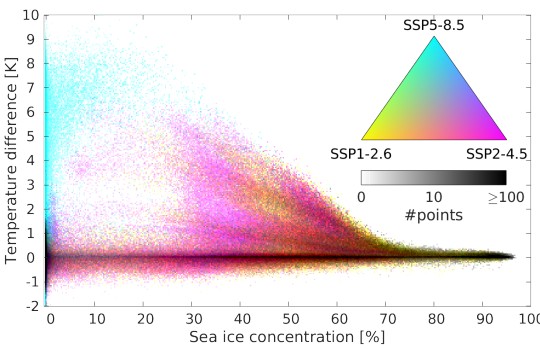

**Figure 8.** Temperature difference between ocean bottom ("benthic" temperature) and the uppermost subsea sediment layer as function of fractional local sea ice concentration for the scenario experiments. Logarithmic density plot (temperature difference in $0.05\ K$ bins, sea ice cover in $0.25\%$ bins) of monthly mean values. The colors are combined RGB-wise according to the color triangle. The color saturation is scaled such that 0 points in a bin corresponds to the value 0 and $\geq 100$ points corresponds to the value 1. Thus for instance a red point contains equally many points from *pmt_ssp126* and *pmt_ssp245* but none from *pmt_ssp585*. Black indicate at least 100 points in all three scenarios. Each scenario contributes more than 460,000 time-space points.

**Table 1.** Performed experiments. The forcing for the three SSPs is identical for the period 1850–2009. $^*$: Forcing repeated cyclicly. "Geoth": Geothermal heat flux included? "Freezing layers": Freezing of porewater allowed for these layers. "Sedi": Sediment heat capacity and conductivity from: "St"andard JSBACH or "Mod"ified as described in the discussion.

| Experiment | Purpose | Period | Forcing | Forcing period | Geoth | Freezing layers | Sedi | Restart @year |
|---|---|---|---|---|---|---|---|---|
| *pmt_ssp126* | main | 1850-3299 | ssp1-2.6 | 1850-3299 | Yes | None | St | |
| *pmt_ssp245* | main | 1850-3309 | ssp2-4.5 | 1850-3309 | Yes | None | St | |
| *pmt_ssp585* | main | 1850-3169 | ssp5-8.5 | 1850-3169 | Yes | None | St | |
| *pmt_pre* | control | 1850-3109 | ssp1-2.6 | 1850-1873$^*$ | Yes | None | St | |
| *pmt_curr* | control | 2010-3109 | ssp1-2.6 | 1986-2009$^*$ | Yes | None | St | pmt_ssp126 2010 |
| *pmt_pre_0* | sensitivity | 1850-3109 | ssp1-2.6 | 1850-1873$^*$ | No | None | St | |
| *pmt_ssp585_0* | sensitivity | 1850-3169 | ssp5-8.5 | 1850-3169 | No | None | St | |
| *pmt_freeze126* | sensitivity | 1850-3169 | ssp1-2.6 | 1850-3169 | Yes | 1–22 (all) | St | |
| *pmt_freeze* | sensitivity | 1850-3169 | ssp5-8.5 | 1850-3169 | Yes | 1–22 (all) | St | |
| *pmt_fr3* | sensitivity | 1850-3169 | ssp5-8.5 | 1850-3169 | Yes | 4–22 | St | |
| *pmt_pre_lc* | sensitivity | 1850-2569 | ssp1-2.6 | 1850-1873$^*$ | Yes | None | Mod | |
| *pmt_ssp585_lc* | sensitivity | 1850-2549 | ssp5-8.5 | 1850-2549 | Yes | None | Mod | |

**Table 2.** Overview over volume (left) and area (right) of SSPF ice in the main and control experiments at selected times. Numbers in brackets indicate relative loss compared to 1850.

| | Volume $10^3\,km^3$ (%) | | | | | Area $10^6\,km^2$ (%) | | | | |
|---|---|---|---|---|---|---|---|---|---|---|
| Year | 1850 | 2100 | 2200 | 2500 | 3000 | 1850 | 2100 | 2200 | 2500 | 3000 |
| Experiment | | | | | | | | | | |
| *pmt_pre* | 153 | 149 (2.1) | 149 ( 2.6) | 147 (4.1) | 143 ( 6.5) | 2.89 | 2.74 ( 5.0) | 2.71 ( 6.1) | 2.68 ( 7.3) | 2.63 ( 8.9) |
| *pmt_curr* | 153 | 149 (2.4) | 148 ( 3.0) | 146 (4.7) | 142 ( 7.3) | 2.89 | 2.72 ( 5.7) | 2.68 ( 7.0) | 2.60 ( 9.9) | 2.55 (11.8) |
| *pmt_ssp126* | 153 | 148 (2.9) | 147 ( 3.9) | 143 (6.1) | 139 ( 9.3) | 2.89 | 2.64 ( 8.5) | 2.53 (12.2) | 2.41 (16.5) | 2.37 (17.8) |
| *pmt_ssp245* | 153 | 148 (3.1) | 145 ( 4.9) | 139 (9.0) | 132 (13.8) | 2.89 | 2.61 ( 9.6) | 2.38 (17.6) | 2.27 (21.3) | 2.22 (23.1) |
| *pmt_ssp585* | 153 | 147 (3.9) | 135 (11.5) | 114 (25.1) | 90 (40.9) | 2.89 | 2.53 (12.3) | 2.21 (23.3) | 2.03 (29.7) | 1.65 (42.7) |