# Peer review of "Strong Increase of Thawing of Subsea Permafrost in the 22nd Century Caused by Anthropogenic Climate Change"

_The Cryosphere, 2021_

## Referee Comment (RC1)

The paper under review is very novel - to the best of my knowledge, this is the first estimate of the SSP scenario-forced sebsea permafrost degradation. However, there are important limitations in the manuscript, which have to be at least properly discussed before final acceptance. Thus, I vote for a major revision for the contemporary version of the manuscript.

**Major comments**

The mentioned in this Section comments are major. However, taking into account the novelty of the paper, I do not insist on changing the setup of simulations or on other time- and/or resource-consuming issues. However, I do guess that the ideas behind these comments have to be thoroughly discussed in the paper or, at least, to be mentioned as limitations of the obtained results.

- The most important concern is for the applied initial conditions. Authors use the SuPerMAP output, which was rerun starting from the LGM conditions to account for transients in the thermal state of the sediments. However, the length of this simulation is likely insufficient to remove the transients completely. In particular, Malakhova and Eliseev (2017, doi 10.1016/j.gloplacha.2017.08.007) showed that at least one complete glacial cycle is needed to equilibrate the thick subsea sediments. Otherwise, the initial condition-forced transients are important even at the depth of several hundred metres below the seafloor - which is certainly of importance for the results of the manuscript. Moreover, in the just mentioned paper as well as in (Malakhova and Eliseev, 2020, doi 10.1016/j.gloplacha.2020.103249), it was shown that the timescale of the permafrost response to external forcing in the middle and outer parts of the Arctice shelf is, at least, 10 kyr or even longer. This response timescale is of the order of the SuPerMAP simulation length, which again indicates possible contamination of the obtained results by the above-mentioned transients.
- Another issue is due to the application of the SuPerMAP output for initial conditions for the JSBACH models. Are the dynamic properties of these two models are sufficiently similar in order not to impose additional transients due to such implicit model change-induced shock?
- The SuPerMAP produces the subsea permafrost even in the areas which are known to lack it. Taking into account applied forcing with a strong benthic warming, this would exaggerate the rate of the permafrost thaw. Additional complications may be due to just mentioned implicit model change.

**Minor comments**

- l. 130: I would suggest to replace Fig. S5 by a figure in which averaging is done only over the permafrost-containing grid cells (coloured area in left panel of Fig. 1) - it would be easier to interpret in the context of the paper.
- ll. 150-165: I am confused with the definition of the permafrost area in the manuscript. Is it a total area of the grid cells which has at least a single frozen layer at some depth? Or authors limit the counting of the frozen layers only up to some (prescribed) depth? Please add the respective explicit definition to the manuscript. Otherwise the quantitative statements on the subsea permafrost area decrease are difficult to interpret.

---

## Author Comment (AC2)

Answer to reviewer comment 2 (by an anonymous reviewer, 2021-10-03)

Over all, the manuscript by Wilkenskjeld et al. is an interesting one describing a first attempt to include benthic sediment temperatures within the framework of an Earth system model across the entire Arctic benthic environment.

We appreciate that the reviewer finds our work interesting and acknowledge the suggestions for improvement. Some of the reviewer's questions are aiming at the workings of the MPI-OM, the oceanic component of MPI-ESM from which we get our most important upper boundary conditions. We need to highlight, that none of the authors have direct experience with this ocean model. Also the experiments from which our data stem (Kleinen et al., 2021) were not conducted with an oceanic focus. We have regrouped some of the questions and comments to improve the text flow.

My main comments are

(1) to try to focus a bit more on the clearly reporting of the results, to show what the model is projecting for each of the scenarios

We will in the revised manuscript further highlight the development and differences of the different scenarios.

(2) to discuss what the initial trend in the benthic temperatures are, and how confident we are in those given the model initialization protocol.

This issue is a major concern for the other reviewer (A. Eliseev). We thus made an elaborate answer to him which we copy here:

The SuPerMAP points were initialized at 50 ka BP with steady state vertical profiles calculated using the geothermal heat flux at 2 km depth and averaged surface temperatures from 400 ka long runs from a limited number points as boundary conditions (see Overduin et al. (2019) for more details). By using these long term averages – longer than the time scale proposed by Malakhova and Eliseev (2017) to be neccesary for the deep sediments to become into equilibrium - this approach rules out transient effects in the deeper parts of the sediments. In the period 50 ka BP to the LGM 23.5 ka BP the SuPerMAP model was run using the boundary conditions as described in Overduin et al (2019). To diminish initial shocks and transient effects from changing the model from SuPerMAP to JSBACH in year 1850, several different ways of doing the transition from the idealized upper boundary conditions described in Overduin et al. (2019) to the "realistic" forcing from MPI-ESM (assuming that the benthic temperatures from the preindustrial period (1850-1873) of the CMIP6 runs are representative for the entire holocene), were tested (Fig. RA1). The approach resulting in the least "noise" turned out to be an interpolation from 23.5 ka BP to 1 ka BP, keeping the forcing constant over the last 1000 years of the SuPerMAP runs. How this interpolative approach exactly compares to the $O(10^4)$ year time scale proposed by Malakhova and Eliseev (2020) for adjusting the sediments to the forcing is not completely clear, but it is at least at the same order of magnitude.

In the manuscript, it may it may not have been made sufficiently clear, that our runs reuse the SuPerMAP runs before the LGM, only modifying the last part of the forcing compared to the experiment presented in Overduin et al. (2019). This may have given rise to the impression that our total spin-up length is only 23.5ka, which would, indeed, limit the usage of our study. We will clarify this in the revised version of the manuscript.

[Figure]

*Fig. RA1: Tested forcing pathways for the SuPerMAP spin-up runs.*

The two models used (SuPerMAP and JSBACH) have very different purposes. Whereas JSBACH is the land component of the MPI-ESM and thus a state-of-the-art terrestrial land and vegetation model with a (normally shallow) soil model designed for typical climate projection time scales of approximately 100-1000 years with high temporal resolution, SuPerMAP is a specialized permafrost, deep soil, model with the capability of repeated ocean transgression/regression cycles, designed for very long time scales. The focus of SuPerMAP is to determine the melting of sub-sea permafrost by geo-thermal heat flux (i.e. from below) and our present work is mainly on the additional melting from above caused by anthropogenic climate change. To be able to extent a very long spin-up to get sensible soil temperature and ice content with climate projections at high temporal resolution (both of forcings and results), this two-model approach has been necessary despite the issues about differences in model physics and interpolation of initial conditions, eventually leading to transient effects.

The two approaches sketched in gray/black colors in Fig. RA1 indeed lead to severe transient effects within the JSBACH runs. In has not been possible to completely eliminate these effects, which are visible as a slightly higher initial melting rate in Figs. 2 and S7 in the manuscript. In Fig. RA2 we used the pmt_pre experiment (which was run with cyclic forcing) to quantify these effects by showing the average melting rate for each for the 24 year forcing cycles. In this experiment a small decrease in Sub-Sea Permafrost (SSPF) ice melting should be expected over time, since individual point may loose all SSPF ice. Despite this small effect, we take the SSPF ice melting from the average from cycle 8 (starting in 2018) to the end of the experiment as the best guess value for the "true" steady state melting rate corresponding to the preindustrial (or holocene) forcing. This average is about 7.4 km$^3$/a. Indeed specially the first cycle (1850-1873) has a melting rate almost twice as high as this average, but already in the second the difference is much diminished (25% additional melting). The average additional melting during the first 7 cycles is about ~1.8 km$^3$/a, only ~25% of the best guess. For comparison the peak melting in pmt_ssp585 is >100 km$^3$/a. Calculating the average melting rates from the pmt_sspXXX experiments for the same 24 year intervals reveals that these exceeds those of pmt_pre already for the third "cycle", i.e. from the year 1898 (not shown), indicating that the climatic signal from here on exceeds the initial "noise". Since our focus is on the inter-SSP differences (after 2010), these transient effects have been considered unimportant, but of course deserve to be mentioned in the discussion in the revised manuscript.

[Figure]

*Fig. RA2: SSPF ice melting in the pmt_pre experiments averaged over each 24 year forcing cycle. Each point mark the average of one forcing cycle. Gray hatch is the mean +/- two standard deviations of the melting from 2018 (forcing cycle 8) to the end of the experiment (to the right of the vertical line).*

While some sensitivity tests were carried out for some of the modelling assumptions made, I am curious whether there are other parameters in the model that could give very different results.

For example, how was the thermal conductivity and porosity of these sediments calculated, and how well constrained are those estimates?

The sediment porosity was adapted from the the SuPerMAP model to keep these two models as consistent as possible. The porosity of SuPerMAP was fitted to the bulk density data of Gu et al. (2014). Also Goto et al. (2017) finds porosity to decrease exponentially with depth.

The thermal properties of our sediment model are those standard values for the JSBACH soil model used for numerous studies including the CMIP6 model comparison project. These are composed of "bedrock" (which in this study is to be interpreted as "sediment") and water fractions weighted by the porosity The values used were: 2 W/Km ("bedrock") and 0.57 W/Km (water) for thermal conductivity and $2*10^6$ J/(Km$^3$) ("bedrock") and $4.218*10^6$ J/(Km$^3$) (water) for heat capacity. We have regarded this part of the model as set and it being a study of itself to adapt it. We thus regarded tuning of these parameters to be beyond the scope of the present study.

Since the question on the sensitivity of our results to these parameters is very relevant, we now conducted additional scenarios (see below), where we replaced the JSBACH "bedrock" values with the those presented in Goto et al. (2017). They report values about 0.8 W/Km and $3.6*10^6$ J/(Km$^3$) for the two parameters respectively from several locations with muddy sediments (porosity >70%) of the coast of Japan. Since JSBACH further adds it's own pore water, we end up with effective values close to those of pure water, which (when no advective processes are accounted for) must be regarded as the most extreme possible setting for delaying heat propagation through the sediments.

Are the timing of results robust with respect to this uncertainty?

Based on the discussion above, we created the new low conductivity scenarios pmt_ssp585_lc and pmt_pre_lc using the values of Goto et al. (2017) and otherwise being identical to pmt_ssp585 and pmt_pre respectively.

Results shows that the time that melting of SSPF ice (Fig. GB3) accelerates is largely unaffected by the change of thermodynamic properties of the sediments. The ratio of the melting to the respective preindustrial scenario (Fig. GB4) is even increasing from about 15 to about 26 in the 22$^{nd}$ century for the low conductivity case, indicating the influence of anthropogenic climate change to be even more dramatic than stated in the manuscript.

[Figure]

*Fig. GB3. Total SSPF ice volume for the standard and "low conductivity" versions of pmt_ssp585 and pmt_pre.*

On the other hand, the absolute melting, is reduced by almost a factor of about 1.8 on average over 1850-2549 in the low conductivity experiments. The absolute melting rate is controlling the change of the IBPT and thus the potential carbon release to the climate system which would thus be accordingly lower. The ratio is higher than average until year 2100 (about 2.5 throughout the 20$^{th}$ century) and generally lower after 2100.

[Figure]

*Fig. GB4. Melting ratios of SSPF ice in pmt_ssp585 and pmt_ssp585lc relative to pmt_pre and pmt_pre_lc respectively.*

Are the heat flux assumptions made in the benthic model consistent with those used in the land model?

The benthic model is an extension of the land model below the ocean, thus using the same heat flux parameterization. We though assume that there is no radiative energy transfer between ocean and sediments.

Also, given the importance of freeze/thaw processes in the manuscript, I'd like to see a bit more description of how the temperature diffusion and latent heat effects were calculated in the model.

Since we did not do any changes to this part of the model, we are hesitating to blow up the size of the paper by copying unchanged material from some of our references. Specifically, details can be found in Ekici et al. (2014) and de Vrese et al. (2020).

Why didn't the authors just set an observed salt gradient rather than disallowing the porewaters to freeze?

This is a nice idea which could have been implemented. It would, however, not take into account the temporal variability (especially: trend) that there might be in the benthic salinity. This could become important when going into the future. For future studies, we will implement a complete salt diffusion scheme which will eliminate the need for the "no freeze" assumption.

Also, what about pressure effects on freezing point depression, it seems like those are relevant to this system?

In the preparations of the SuPerMAP runs, which served as initial conditions for the present study, hydrostatic pressure effects on freezing point depression were considered. Preliminary calculations revealed that the effect of pressure on the freezing point depression is small relative to that of dissolved salts, especially over the depth ranges of interest for permafrost (few hundreds of meters). It was therefore not included in the SuPerMAP calculations (Overduin et al. (2019)). Including it in the present study would therefore enhance the differences between the two models and thus lead to an increased initial shock in our model.

Including hydrostatic pressure effects would e.g. at 500 m depth lower the freezing point by around 0.35°C and thus tend to lower the amount of ice in the lower portion of modelled permafrost. Since the majority of the effects discussed in the present study are much further up in the sediments (<100 m depth) the effects will be proportionally smaller.

I see that there is some discussion of the effects of this assumption in section 4.2, but I think some clearer discussion of this here would help.

We will extend and clarify the discussion on this point in the revised manuscript.

Also what are the nominal resolutions of the GR30 and T63 grids?

T63 is a spherical grid (equivalent to a lat/lon grid with small variations in the distance between the latitudes – specially very close to the poles) with a resolution of about 1.9 x 1.9 degrees all over the globe. At the latitudes of Arctic SSPF this corresponts to approximately 50 x 200 km as pointed out in line 80 of the manuscript. The GR30 is a lat/lon-grid with rotated poles (placed on Greenland and somewhere on the Antarctic continent) which makes it difficult to assign it with a nominal resolution. In the relevant latitudes (65N through 80N) the average point distance is about 70 km, ranging from about 40 km near Greenland to about 200 km off the Siberian coast (Fig. RB5).

[Figure]

*Fig. RB5: Distribution of MPI-OM GR30 points between 65N and 80N.*

What is the depth distribution of ocean model layers?

The MPI-OM setup uses 40 layers a fixed depths. The layer midpoints in meters are:

7.5, 20, 30, 40, 50, 60, 71.5, 85.5, 103, 125.5, 153, 185.5, 223, 265.5, 313, 365.5, 423, 488, 563, 648, 743, 848, 963, 1088, 1223, 1368, 1528, 1703, 1888, 2083, 2293, 2528, 2788, 3073, 3398, 3773, 4198, 4673, 5173, 5723.

p. 2, section 2.1: Some more info is needed on how MPI-OM calculates the benthic temperature.

The benthic temperature is taken as the temperature of the lowest oceanic model layer above the ocean bottom. For further infomation here we could only copy from published literature on MPI-OM (e.g. Mauritzen et al. (2019), Jungclaus et al. (2006, 2013)).

How well does the model represent the observed benthic climate?

Referring to the observations presented in Dmitrenko et al. (2011) in the Laptev Sea, reporting an increase of benthic summer temperatures of 2.1 K between 1985 and 2009, we looked into similar area and period in our data (Fig. RB6 and RB7).

[Figure]

*Fig. RB6: Change in benthic summer temperature from 1985 to 2009 (all scenarios except pmt_pre). Cyan dots mark the 11 points (Laptev and eastern Kara Seas) included in the time series in Fig. 2. These 11 points also happens to be those with the highest temperature change.*

[Figure]

Fig RB7: Time series of benthic summer (JAS) temperatures for points marked in Fig. RB6 (all scenarios except pmt_pre). The difference between 2009 and 1985 (red dots) is ~3.5K and the linear trend (red line) indicate a warming of 2.15K in this period. This is in very close agreement with the results of Dmitrenko et al. (2011).

Based on this analysis we conclude that:

1) The GR30 version of MPI-OM seems to be doing a remarkable good job in reproducing the benthic temperature changes on the shelves of the Laptev and Kara seas, despite its coarse resolution in this area (see above).

2) The warming reported in Dmitrenko et al. (2011) seems to be a regional phenomenon and – looking at the time series – to be limited to these few decades. As well the minimum and maximum of the period between 1985 and 2009 are within the range of the temperature variations between 1850 and 1984.

Based on this case study, which seems to be a rather extreme case, we expect the modelled benthic temperatures to be roughly in agreement with observations. Also it seems plausible that the development of ocean temperatures (including the benthic) on the shelves roughly follows that of the atmospheric temperatures – eventually with some delay. This effect is for the future part of our study the most important.

Also what aspects of the CLIMBER output were used to force MPI-ESM, just CO2 and other GHGs?

Only $CO_2$ was adopted from the CLIMBER simulations as boundary conditions for the MPI-ESM runs creating our benthic temperatures. $CH_4$ were calculated interactively in MPI-ESM and $N_2O$ was kept constant. See Kleinen et al. (2021) for details.

p. 3 l. 70-79. These two paragraphs seem in conflict with each other. Either salt diffusion is slow and unimportant, or it is fast and important.

Our assumption with respect to salt diffusion is that it is too slow to be important at the time scale we cover in the present study (O(1000yr)), though it is definitely play an important role for longer (i.e. glacial) time scales. We will clarify this in the revised manuscript.

p. 5, l. 136 Is this thawing under preindustrial forcing to be interpreted as lagged thaw following LGM, or is it an artifact of the imposed initial conditions?

The SSPF has since it's inundation generally been overlaid by comparatively warm oceanic water, causing a steady thawing from above which – at some point in time – will cause all SSPF to be melted away (provided that the world does not enter a new glacial period causing an oceanic regression from the SSPF areas on the oceanic shelves). Therefore the thaw seen applying preindustrial forcing is a lagging effect of the changed climatic conditions since the LGM. Only exception is – as discussed above – a minor additional thaw in the period ~1850-1900 which is either caused by inconsistencies between the physics of SuPerMAP and JSBACH or by physical inconsistencies in the initial conditions of JSBACH arising from the interpolation between the vertical grids of the two models.

Since the general fate of SSPF (independently of climate change) is the basic setting of our study, we will extend the discussion and explanation in the introduction.

figs. 1, 5, &6: Is 'depth' the depth below the benthic surface or the depth below the sea surface?

Since we haven't dealt with the ocean model ourselves, we do not at all consider the ocean as part of our model domain. Therefore we're always having depth as "below the benthic surface/ocean bottom", which is at a fixed depth since we did not include sedimentation in the JSBACH model. We will clarify this in the revised manuscript.

fig. 3: Is the ratio of melt rate to the preindustrial melt rate a meaningful metric, and if so why? It seems like the absolute loss rate is a more fundamental measure than the relative rate. But if the relative rate is more meaningful, then some background and explanation would be helpful.

We regard both the relative ratio (Fig. 3 in the paper and Fig. GB4) and the absolute melting (Fig. 2 in the paper and Fig. GB3) as important. The relative ratio highlights the influence of the climatic changes, since – as described above – there's always a "background" melting caused by the imbalance remaining from post-LGM climate change and inundation. This is even more highlighted by the additional scenarios pmt_pre_lc and pmt_ssp585_lc (above). Since the absolute melting via the IBPT controls the release rate of carbon (mainly $CH_4$) to the ocean, one could phrase it as: The absolute melting states how much damage is done by the melting of SSPF and the relative ratio states how much of this damage is due to (anthropogenic) climate change.

fig. 4: It would be helpful to see some time progression here. For example how much ice will have melted by 2100, 2200, 2300, 2500, and 3000 under each of the scenarios?

The patterns of melting do not change much over time. Therefore we considered these interstadials as being uninteresting. However, taking into account the focus of the paper on the earlier part of the time period (say: before year 2400), it might be worth considering putting a picture of e.g. 2300 into the main text. Additional time slices we can put in the supplement.

fig. 7: Are these annual mean values, and if so, should these be thought of as the spatial area with no ice, or the fraction of the year with no ice?

These are monthly mean values, which are averages of the all the instantaneous (i.e. from each MPI-OM time step) fractional sea ice concentrations for the individual months. Thus it is a combination of an instantaneous fraction (i.e. spatial mean) and a temporal mean.

fig. 7: Why the break in slope at 70% sea ice concentration?

This is hard to say in detail, since this figure basically shows only a correlation, not a causality. In MPI-OM, the sea ice concentration is assumed to be uniformly distributed over the the grid cell. This has the effect, that all sea ice has to melt away before energy is available for heating the ocean water. Our best guess is that for sea ice concentrations >70% sea ice is present for such a long time (within the month) that there's no time to heat the water. Also oceanic advection from neighbouring grid cells may play a role. Anyway: It seems obvious that there has to be a point where the insulating role of sea ice start vanishing, but we have no good explanation why it specifically occurs at around 70%.

fig. 7: Would the plot look different if other explanatory variables (e.g. surface ocean temperature or surface air temperature) were used?

Since the benthic temperatures are the direct driver of the energy input into the sediments we expect a much more clear picture using this variable. The SST is – though it is very similar to the benthic temperatures except in late summer – not quite an as direct driver, and surface air temperature is even less so. A spatially limited illustration of the relation to the surface air temperature can be seen in Fig. 6 of the manuscript (lower panel). This clearly shows the decoupling of the warming of the lower atmosphere and the sediments. For a complete picture we have re-plotted Fig. 7 of the manuscript as function of SST (Fig. GB8 left) and surface air temperature (Fig. GB8 right) instead of benthic temperature. These pictures do in our minds not show any clear interesting patterns.

[Figure]

*Fig. GB8. As Fig. 7 in the manuscript, but replacing the benthic temperature with SST (left) and surface air temperature (right) respectively.*

p. 8, Line 244: does this loss also occur under a steady-state preindustrial climate, or is it a forced response to the historical warming?

The experiment with preindustrial forcing losses about 76% of the SSPF area mentioned in the paper for the historical experiments (Fig. GB9). We attribute this initial loss mainly to loss of thin SSPF ice in the upper layers (which may in parts be an unrealistic artifact of the initialization) for which also the preindustrial forcing is sufficient to melt away. Also the transient effects of the model change discussed above may play a role here. The remaining 24% is due to warming during the historical period. This interesting point will be discussed in the revised manuscript.

[Figure]

*Fig. GB9. Pan-arctic area with SSPF and it's development over time for the scenarios presented in the manuscript. The discrete steps are due to the grid cell discretization. Total modelled area is about 4.6 mill. $km^2$ distributed on 407 points which each either have or does not have SSPF at a certain time.*

The unit somehow disappeared from this figure, sorry. It is mK/m (or K/km). Will of course be corrected in the revised manuscript.

fig. S6 Is helpful in interpreting the results, I suggest moving to the main manuscript document.

This figure shows our forcing data only, which we didn't produce ourselves. We judged that we by putting it into the main manuscript risk to confuse what is actually our work and what we included from the work of others. However, since it seems to be more important for the interpretation than anticipated, we will incorporate it in the main part of the revised manuscript.

fig. S11, the right hand side panel isn't interpretable, I think you'd need to show the amplitude for each scenario separately, and using the same color scale for both the historical and future panels.

We will redo this figure for the revised manuscript, separating the three scenarios.

Bibliography

de Vrese, P., Stacke, T., Kleinen, T., and Brovkin, V.: Diverging responses of high-latitude CO 2 and CH 4 emissions in idealized climate change scenarios, The Cryosphere, 15, 1097–1130, https://doi.org/10.5194/tc-15-1097-2021, 2021.

Ekici, A., Beer, C., Hagemann, S., Boike, J., Langer, M., and Hauck, C.: Simulating high-latitude permafrost regions by the JSBACH terrestrial ecosystem model, Geoscientific Model Development, 7, 631–647, https://doi.org/10.5194/gmd-7-631-2014, 2014.

Goto, S., Yamano, M., Morita, S., Kanamatsu, T., Hachikubo, A., Kataoka, S., Tanahashi, M., and Matsumoto, R.: Physical and thermal properties of mud-dominant sediment from the Joetsu Basin in the eastern margin of the Japan Sea, Marine Geophysical Research, 38, 393–407, https://doi.org/10.1007/s11001-017-9302-y, 2017.

Gu, X., Tenzer, R., and Gladkikh, V.: Empirical models of the ocean-sediment and marine sediment-bedrock density contrasts, Geosciences Journal, 18, 439–447, https://doi.org/10.1007/s12303-014-0015-9, 2014.

Jungclaus, J. H., Fischer, N., Haak, H., Lohmann, K., Marotzke, J., Matei, D., Mikolajewicz, U., Notz, D., and von Storch, J. S.: Characteristics of the ocean simulations in the Max Planck Institute Ocean Model (MPIOM) the ocean component of the MPI-Earth system model, Journal of Advances in Modeling Earth Systems, 5, 422–446, https://doi.org/https://doi.org/10.1002/jame.20023, 2013.

Jungclaus, J. H., Keenlyside, N., Botzet, M., Haak, H., Luo, J. J., Latif, M., Marotzke, J., Mikolajewicz, U., and Roeckner, E.: Ocean circulation and tropical variability in the coupled model ECHAM5/MPI-OM, JOURNAL OF CLIMATE, 19, 3952–3972, https://doi.org/10.1175/JCLI3827.1, 2006.

Kleinen, T., Gromov, S., Steil, B., and Brovkin, V.: Atmospheric methane underestimated in future climate projections, ENVIRONMENTAL RESEARCH LETTERS, 16, https://doi.org/10.1088/1748-9326/ac1814, 2021.

Mauritsen, T., Bader, J., Becker, T., Behrens, J., Bittner, M., Brokopf, R., Brovkin, V., Claussen, M., Crueger, T., Esch, M., Fast, I., Fiedler, S., Flaeschner, D., Gayler, V., Giorgetta, M., Goll, D. S., Haak, H., Hagemann, S., Hedemann, C., Hohenegger, C., Ilyina, T., Jahns, T., Jimenez-de-la Cuesta, D., Jungclaus, J., Kleinen, T., Kloster, S., Kracher, D., Kinne, S., Kleberg, D., Lasslop, G., Kornblueh, L., Marotzke, J., Matei, D., Meraner, K., Mikolajewicz, U., Modali, K., Moebis, B., Muellner, W. A., Nabel, J. E. M. S., Nam, C. C. W., Notz, D., Nyawira, S.-S., Paulsen, H., Peters, K., Pincus, R., Pohlmann, H., Pongratz, J., Popp, M., Raddatz, T. J., Rast, S., Redler, R., Reick, C. H., Rohrschneider, T., Schemann, V., Schmidt, H., Schnur, R., Schulzweida, U., Six, K. D., Stein, L., Stemmler, I., Stevens, B., von Storch, J.-S., Tian, F., Voigt, A., Vrese, P., Wieners, K.-H., Wilkenskjeld, S., Winkler, A., and Roeckner, E.: Developments in the MPI-M Earth System Model version 1.2 (MPI-ESM1.2) and Its Response to Increasing CO2, JOURNAL OF ADVANCES IN MODELING EARTH SYSTEMS, 11, 998–1038, https://doi.org/10.1029/2018MS001400, 2019.

---

## Author Response (AR1)

Answer to reviewer comments 1 (by Alexey V. Eliseev, 2021-09-16)

We, the authors, of „Strong Increase of Thawing of Subsea Permafrost in the 22nd Century Caused by Anthropogenic Climate Change" appreciate very much the acknowledgement of the novelty of our work by the reviewer and the fast reply. Below we attempt to answer the points of critics from the reviewer more or less point by point.

**Major comments:**

*The reviewer points out that our spin-up procedure, change of models (from SuPerMAP to JSBACH) at the year 1850 and thereby resultant transient model drifts may limit the value of our results.*

[2nd paragraph of sec. 2.5 (Initial conditions) essentially rewrote and extended.] The SuPerMAP points were initialized at 50 ka BP with steady state vertical profiles calculated using the geothermal heat flux at 2 km depth and averaged surface temperatures from 400 ka long runs from a limited number points as boundary conditions (see Overduin et al. (2019) for more details). By using these long term averages – longer than the time scale proposed by Malakhova and Eliseev (2017) to be neccesary for the deep sediments to become into equilibrium - this approach rules out transient effects in the deeper parts of the sediments. In the period 50 ka BP to the LGM 23.5 ka BP the SuPerMAP model was run using the boundary conditions as described in Overduin et al (2019). To diminish initial shocks and transient effects from changing the model from SuPerMAP to JSBACH in year 1850, several different ways of doing the transistion from the idealized upper boundary conditions described in Overduin et al. (2019) to the "realistic" forcing from MPI-ESM (assuming that the benthic temperatures from the preindustrial period (1850-1873) of the CMIP6 runs are representative for the entire holocene), were tested (Fig. RA1). The one resulting in the least "noise" turn out to be an interpolation from 23.5 ka BP to 1 ka BP, keeping the forcing constant over the last 1000 years of the SuPerMAP runs. How this interpolative approach exactly compares to the $O(10^4)$ year time scale suggested in Malakhova and Eliseev (2020) for adjusting the sediments to the forcing is not completely clear, but at least at the same order of magnitude.
In the paper, it may it may not have been made sufficiently clear, that our runs reuse the SuPerMAP runs before the LGM, only modifying the last part of the forcing compared to the experiment presented in Overduin et al. (2019). This may have given rise to the impression that our total spin-up length is only 23.5ka, which would, indeed, limit the usage of our study. We will clarify this in the revised version of the manuscript.

[Figure]

*Fig. RA1: Tested forcing pathways for the SuPerMAP spin-up runs.*

The two models used (SuPerMAP and JSBACH) have very different purposes. Whereas JSBACH is the land component of the MPI-ESM and thus a state-of-the-art terrestrial land and vegetation model with a (normally shallow) soil model designed for typical climate projection time scales of approximately 100-1000 years with high temporal resolution, SuPerMAP is a specialized

permafrost, deep soil, model with the capability of repeated ocean transgression/regression cycles, designed for very long time scales. The focus of SuPerMAP is to determine the melting of sub-sea permafrost by geo-thermal heat flux (i.e. from below) and our present work is mainly on the additional melting from above caused by anthropogenic climate change. To be able to extent a very long spin-up to get sensible soil temperature and ice content with climate projections at high temporal resolution (both of forcings and results), this two-model approach has been necessary despite the issues about differences in model physics and interpolation of initial conditions, eventually leading to transient effects.

The two approaches sketched in gray/black colors in Fig. RA1 indeed lead to severe transient effects within the JSBACH runs. In has not been possible to completely eliminate these effects, which are visible as a slightly higher initial melting rate in Figs. 2 and S7 in the paper. In Fig. RA2 we used the pmt_pre experiment (which was run with cyclic forcing) to quantify these effects by showing the average melting rate for each for the 24 year forcing cycles. Though a small decrease in Sub-Sea Permafrost (SSPF) ice melting should be expected over time individual point may loose all SSPF ice, we take the SSPF ice melting from the average from cycle 8 (starting in 2018) to the end of the experiment as the best guess value for the "true" steady state melting rate corresponding to the preindustrial (or holocene) forcing. This average is about 7.4 km$^3$/a. Indeed specially the first cycle (1850-1873) has a melting rate almost zwice as high as this average, but already in the second the difference is much diminished (25% additional melting). In total the additional melting in the first 7 cycles is about ~1.8 km$^3$/a only 25% on top of the best guess. For comparison the peak melting in pmt_ssp585 is >100 km$^3$/a. Calculating the average melting rates from the pmt_sspXXX experiments for the same 24 year intervals reveals that these exceeds those of pmt_pre already for the third "cycle", i.e. from the year 1898 (not shown), indicating that the climatic signal from here on exceeds the initial "noise". Since our focus is on the inter-SSP differences (after 2010), these transient effects have been considered unimportant, but of course deserve to be mentioned in the discussion in the revised manuscript.

[Figure]

Fig. RA2: SSPF ice melting in the pmt_pre experiments averaged over each 24 forcing cycle. Each point mark the average of one forcing cycle. Gray hatch is the mean +/- two standard deviations of the melting from 2018 (forcing cycle 8) to the end of the experiment (to the right of the vertical line).

*The reviewer has the opinion that SuPerMAP produces too much SSPF ice.*

[Discussion added to sec. 4.3 (Area of SSPF).] It is true that SuPerMAP produces subsea permafrost in some regions which probably lack it. However, it is important here to be specific about the nuances of terminology. The permafrost definition used in the SuPerMAP study corresponds to terrestrial permafrost (cryotic for more than 2 consecutive years) and will therefore cover a larger spatial region than permafrost defined by ice content. This issue is addressed in the Overduin et al. (2019) ("*...Both effects lead to an overestimation of the areal extent of cryotic sediments...*"). The validation data sets for SuPerMAP nonetheless fit observed data in Alaska and the Kara Sea reasonably well at the scale of the modelling. These data sets are based on seismic velocity and therefore ice content, suggesting that discrepancies between defined permafrost

domains are small and/or site specific. For example, SuPerMAP underestimates permafrost determined by borehole investigations in the Canadian Beaufort, where ice cover histories and the influence of fluvial water may be poorly constrained. Strong benthic warming might exaggerate the speed with which permafrost reaches its thawing temperature, but the rate of permafrost thaw is more strongly dependent on sediment ice content. This is unlikely to be relevant for permafrost that is cryotic but ice poor.

**Minor comments:**

*The reviewer suggest for clarity to show the temporal development of the benthic temperature only for points with SSPF in Fig. S5.*

[Figure modified and added to the main manuscript (new fig. 2)] We acknowledge that this is a good idea. The resultant figure is shown here as Fig. RA3:

[Figure]

*Fig. RA3: Benthic temperature development. As supplemental figure S5 but average only done over points with SSPF.*

*The reviewer miss a clear definition of which area is defined as SSPF area.*

[Definition added in last paragraph of sec. 2.5 (Initial conditions)] In our study we regard a grid cell with an ice concentration >0 anyhere in the sediment column as a SSPF cell, which is counted with it's total area in the SSPF area. We have an implicit depth limit by the limit of our sediment depth (1km) but no futher limit on the depth range in which ice has to appear to be counted. Due to the geothermal heat flux and the freezing history anyhow, no ice is found below 700m (i.e. in our lowest layer). We will include a more accurate definition in the final manuscript.

Bibliography

Malakhova, V. V. and Eliseev, A. V.: The role of heat transfer time scale in the evolution of the subsea permafrost and associated methane hydrates stability zone during glacial cycles, GLOBAL AND PLANETARY CHANGE, 157, 18–25, 375 https://doi.org/10.1016/j.gloplacha.2017.08.007, 2017.

Malakhova, V. V. and Eliseev, V, A.: Uncertainty in temperature and sea level datasets for the Pleistocene glacial cycles: Implications for thermal state of the subsea sediments, GLOBAL AND PLANETARY CHANGE, 192, https://doi.org/10.1016/j.gloplacha.2020.103249, 2020.

Overduin, P. P., Schneider von Deimling, T., Miesner, F., Grigoriev, M. N., Ruppel, C., Vasiliev, A., Lantuit, H., Juhls, B., and Westermann, S.: Submarine Permafrost Map in the Arctic Modeled Using

1-D Transient Heat Flux (SuPerMAP), Journal of Geophysical Research: Oceans, 0, https://doi.org/ 10.1029/2018JC014675, 2019.

Answer to anonymous reviewer #2

Over all, the manuscript by Wilkenskjeld et al. is an interesting one describing a first attempt to include benthic sediment temperatures within the framework of an Earth system model across the entire Arctic benthic environment.

We appreciate that the reviewer finds our work interesting and acknowledge the suggestions for improvement. Some of the reviewer's questions are aiming at the workings of the MPI-OM, the oceanic component of MPI-ESM from which we get our most important upper boundary conditions. We need to highlight, that none of the authors have direct experience with this ocean model. Also the experiments from which our data stem (Kleinen et al., 2021) were not conducted with an oceanic focus. We have regrouped some of the questions and comments to improve the text flow.

My main comments are

(1) to try to focus a bit more on the clearly reporting of the results, to show what the model is projecting for each of the scenarios

[We have at various points tried to clarify what belongs to which scenario. Since, however, we found the results already well separated, this diffuse comment is a bit puzzling.] We will in the revised manuscript further highlight the development and differences of the different scenarios.

(2) to discuss what the initial trend in the benthic temperatures are, and how confident we are in those given the model initialization protocol.

This issue is a major concern for the other reviewer (A. Eliseev). We thus made an elaborate answer to him which we copy here:

[2nd paragraph of sec. 2.5 (Initial conditions) essentially rewrote and extended.] The SuPerMAP points were initialized at 50 ka BP with steady state vertical profiles calculated using the geothermal heat flux at 2 km depth and averaged surface temperatures from 400 ka long runs from a limited number points as boundary conditions (see Overduin et al. (2019) for more details). By using these long term averages – longer than the time scale proposed by Malakhova and Eliseev (2017) to be neccesary for the deep sediments to become into equilibrium - this approach rules out transient effects in the deeper parts of the sediments. In the period 50 ka BP to the LGM 23.5 ka BP the SuPerMAP model was run using the boundary conditions as described in Overduin et al (2019). To diminish initial shocks and transient effects from changing the model from SuPerMAP to JSBACH in year 1850, several different ways of doing the transistion from the idealized upper boundary conditions described in Overduin et al. (2019) to the "realistic" forcing from MPI-ESM (assuming that the benthic temperatures from the preindustrial period (1850-1873) of the CMIP6 runs are representative for the entire holocene), were tested (Fig. RB1). The one resulting in the least "noise" turn out to be an interpolation from 23.5 ka BP to 1 ka BP, keeping the forcing constant over the last 1000 years of the SuPerMAP runs. How this interpolative approach exactly compares to the $O(10^4)$ year time scale suggested in Malakhova and Eliseev (2020) for adjusting the sediments to the forcing is not completely clear, but at least at the same order of magnitude.
In the paper, it may it may not have been made sufficiently clear, that our runs reuse the SuPerMAP runs before the LGM, only modifying the last part of the forcing compared to the experiment presented in Overduin et al. (2019). This may have given rise to the impression that our total spin-up length is only 23.5ka, which would, indeed, limit the usage of our study. We will clarify this in the revised version of the manuscript.

[Figure]

*Fig. RB1: Tested forcing pathways for the SuPerMAP spin-up runs.*

The two models used (SuPerMAP and JSBACH) have very different purposes. Whereas JSBACH is the land component of the MPI-ESM and thus a state-of-the-art terrestrial land and vegetation model with a (normally shallow) soil model designed for typical climate projection time scales of approximately 100-1000 years with high temporal resolution, SuPerMAP is a specialized permafrost, deep soil, model with the capability of repeated ocean transgression/regression cycles, designed for very long time scales. The focus of SuPerMAP is to determine the melting of sub-sea permafrost by geo-thermal heat flux (i.e. from below) and our present work is mainly on the additional melting from above caused by anthropogenic climate change. To be able to extent a very long spin-up to get sensible soil temperature and ice content with climate projections at high temporal resolution (both of forcings and results), this two-model approach has been necessary despite the issues about differences in model physics and interpolation of initial conditions, eventually leading to transient effects.

The two approaches sketched in gray/black colors in Fig. RB1 indeed lead to severe transient effects within the JSBACH runs. In has not been possible to completely eliminate these effects, which are visible as a slightly higher initial melting rate in Figs. 2 and S7 in the paper. In Fig. RB2 we used the pmt_pre experiment (which was run with cyclic forcing) to quantify these effects by showing the average melting rate for each for the 24 year forcing cycles. Though a small decrease in Sub-Sea Permafrost (SSPF) ice melting should be expected over time individual point may loose all SSPF ice, we take the SSPF ice melting from the average from cycle 8 (starting in 2018) to the end of the experiment as the best guess value for the "true" steady state melting rate corresponding to the preindustrial (or holocene) forcing. This average is about 7.4 km$^3$/a. Indeed specially the first cycle (1850-1873) has a melting rate almost zwice as high as this average, but already in the second the difference is much diminished (25% additional melting). In total the additional melting in the first 7 cycles is about ~1.8 km$^3$/a only 25% on top of the best guess. For comparison the peak melting in pmt_ssp585 is >100 km$^3$/a. Calculating the average melting rates from the pmt_sspXXX experiments for the same 24 year intervals reveals that these exceeds those of pmt_pre already for the third "cycle", i.e. from the year 1898 (not shown), indicating that the climatic signal from here on exceeds the initial "noise". Since our focus is on the inter-SSP differences (after 2010), these transient effects have been considered unimportant, but of course deserve to be mentioned in the discussion in the revised manuscript.

[Figure]

*Fig. RB2: SSPF ice melting in the pmt_pre experiments averaged over each 24 forcing cycle. Each point mark the average of one forcing cycle. Gray hatch is the mean +/- two standard deviations of the melting from 2018 (forcing cycle 8) to the end of the experiment (to the right of the vertical line).*

While some sensitivity tests were carried out for some of the modelling assumptions made, I am curious whether there are other parameters in the model that could give very different results.

For example, how was the thermal conductivity and porosity of these sediments calculated, and how well

constrained are those estimates?

[Think this is already explained in sufficient detail in the manuscript.] The sediment porosity was adapted from the the SuPerMAP model to keep these two models as consistent as possible. The porosity of SuPerMAP was fitted to the bulk density data of Gu et al. (2014). Also Goto et al. (2017) finds porosity to decrease exponentially with depth.

[Paragraph added to sec. 4.2 (Model limitations and assumptions).] The thermal properties of our sediment model are those standard values for the JSBACH soil model used for numerous studies including the CMIP6 model comparison project. These are composed of porosity weighted "bedrock" (which in this study is to be interpreted as "sediment") and water fractions with the values: 2 W/Km ("bedrock") and 0.57 W/Km (water) for thermal conductivity and $2*10^6$ J/(Km$^3$) ("bedrock") and $4.218*10^6$ J/(Km$^3$) (water) for heat capacity. We have regarded this part of the model as set and it being a study of itself to adapt it. We thus regarded tuning of these parameters to be beyond the scope of the present study.

Since the question on the sensitivity of our results to these parameters is very relevant, we conducted additional scenarios (see below), where we replaced the JSBACH "bedrock" values with the those presented in Goto et al. (2017). They report values about 0.8 W/Km and $3.6*10^6$ J/(Km$^3$) for the two parameters respectively from several locations with muddy sediments (porosity >70%) of the coast of Japan. Since JSBACH further adds it's own pore water, we end up with effective values close to those of pure water, which (when no advective processes are accounted for) must be regarded as the most extreme possible setting for delaying heat propagation through the sediments.

Are the timing of results robust with respect to this uncertainty?

[See above.] Based on the discussion above, we created the new scenarios pmt_ssp585_lc and pmt_pre_lc using the values of Goto et al. (2017) and otherwise being identical to pmt_ssp585 and pmt_pre respectively.

Results shows that the time that melting of SSPF ice (Fig. GB3) accelerates is largely unaffected by the change of thermodynamic properties of the sediments. The ratio of the melting to the respective preindustrial scenario (Fig. GB4) is even increasing from about 15 to about 26 in the 22nd century for the low conductivity case, indicating the influence of anthropogenic climate change to be even more dramatic than stated in the paper draft.

On the other hand, the absolute melting, is reduced by almost a factor of about 1.8 on average over 1850-2549 in the low conductivity experiments. The absolute melting rate is controlling the change of the IBPT and thus the potential carbon release to the climate system which would thus be accordingly lower. The ratio is higher than average until year 2100 (about 2.5 throughout the 20th century) and generally lower after 2100.

[Figure]

*Fig. GB3. Total SSPF ice volume for the standard and "low conductivity" versions of pmt_ssp585 and pmt_pre.*

*Fig. GB4. Melting ratios of SSPF ice in pmt_ssp585 and pmt_ssp585lc relative to pmt_pre and pmt_pre_lc respectively.*

Are the heat flux assumptions made in the benthic model consistent with those used in the land model?

[The "no change" wrt. Std. Model is emphasized in sec. 2.4 (Model setup and experiments).] The benthic model is an extension of the land model below the ocean, thus using the same heat flux parameterization. We though assume that there is no radiative energy transfer between ocean and sediments.

Also, given the importance of freeze/thaw processes in the manuscript, I'd like to see a bit more description of how the temperature diffusion and latent heat effects were calculated in the model.

[Refrences repeated to sec. 2.4 (Model setup and experiments).] Since we did not do any changes to this part of the model, we are hesitating to blow up the size of the paper by copying unchanged material from some of our references. Specifically, details can be found in Ekici et al. (2014) and de Vrese et al. (2020).

Why didn't the authors just set an observed salt gradient rather than disallowing the porewaters to freeze?

[Don't think it is appropriate to discuss alternative speculative assumptions in the manuscript] This is a nice idea which could have been implemented. It would, however, not take into account the temporal variability (especially: trend) that there might be in the benthic salinity. This could become important when going into the future. For future studies, we will implement a complete salt diffusion scheme which will eliminate the need for the "no freeze" assumption.
Also, what about pressure effects on freezing point depression, it seems like those are relevant to this system?

[Paragraph added to sec. 4.2 (Model limitations and assumptions).] In the preparations of the SuPerMAP runs, which served as initial conditions for the present study, hydrostatic pressure effects on freezing point were depression considered. Preliminary calculations revealed that the effect of pressure on the freezing point depression is small relative to that of dissolved salts, especially over the depth ranges of interest for permafrost (few hundreds of meters). It was therefore not included in the SuPerMAP calculations. Including it in the present study would therefore enhance the differences between the two models and thus lead to an increased initial shock in our model.

Including hydrostatic pressure effects would e.g. at 500 m depth lower the freezing point by around 0.35°C and thus tend to lower the amount of ice in the lower portion of modelled permafrost. Since the majority of the effects discussed in the present study are much higher in the sediments (<100 m depth) the effects will proportionally smaller.

I see that there is some discussion of the effects of this assumption in section 4.2, but I think some clearer discussion of this here would help.

[See above] We will extend and clarify the discussion on this point in the revised manuscript.

Also what are the nominal resolutions of the GR30 and T63 grids?

[Added comment on GR30 resolution to sec. 2.6 (Boundary conditions).] T63 is a spherical grid with a resolution of about 1.9 x 1.9 degrees all over the globe. At the latitudes of Arctic SSPF this corresponds to approximately 50 x 200 km as pointed out in line 80 of the manuscript. The GR30 is a lat/lon-grid with rotated poles (placed on Greenland and somewhere on the Antarctic continent) which makes it difficult to assign it with a nominal resolution. In the relevant latitudes (65N through 80N) the average point distance is about 70 km, ranging from about 40 km near Greenland to about 200 km off the Siberian coast (Fig. RB5).

*Fig. RB5: Distribution of MPI-OM GR30 points between 65N and 80N.*

[Figure]

What is the depth distribution of ocean model layers?

[This is too detailed for the manuscript except the #layers (now in sec. 2.6 (Boundary conditions)).]The MPI-OM setup uses 40 layers a fixed depths. The layer midpoints in meters are:

7.5, 20, 30, 40, 50, 60, 71.5, 85.5, 103, 125.5, 153, 185.5, 223, 265.5, 313, 365.5, 423, 488, 563, 648, 743, 848, 963, 1088, 1223, 1368, 1528, 1703, 1888, 2083, 2293, 2528, 2788, 3073, 3398, 3773, 4198, 4673, 5173, 5723

p. 2, section 2.1: Some more info is needed on how MPI-OM calculates the benthic temperature.

[Refrences added to sec. 2.1 (Submerging) and description of "benthic temperatures" to sec. 2.6 (Boundary conditions).] The benthic temperature is taken as the temperature of the lowest oceanic model layer above the ocean bottom. For further infomation here we could only copy from published literature on MPI-OM (e.g. Mauritzen et al. (2019), Jungclaus et al. (2006, 2013)).

How well does the model represent the observed benthic climate?

[Two sentences added to sec. 2.6 (Boundary conditions).] Referring to the observations presented in Dmitrenko et al. (2011) in the Laptev Sea, reporting an increase of benthic summer temperatures of 2.1 K between 1985 and 2009, we looked into similar area and period in our data (Fig. RB6 and RB7).

[Figure]

*Fig. RB6: Change in benthic summer temperature from 1985 to 2009 (all scenarios except pmt_pre). Cyan dots mark the 11 points (Laptev and eastern Kara Seas) include in the time series in Fig. 2. These 11 points also happens to be those with the highest temperature change.*

[Figure]

Fig RB7: Time series of benthic summer (JAS) temperatures for points marked in Fig. 1 (all scenarios except pmt_pre). The difference between 2009 and 1985 (red dots) is ~3.5K and the linear trend (red line) indicate a warming of 2.15K in the period. This is in very close agreement with the results of Dmitrenko et al. (2011).

Based on this analysis we conclude that:

1) The GR30 version of MPI-OM seems to be doing a remarkable good job in reproducing the benthic temperature changes on the shelves of the Laptev and Kara seas, despite its coarse resolution in this area (see above).

2) The warming reported in Dmitrenko et al. (2011) seems to be a regional phenomenon and – looking at the time series – to be limited to these few decades. As well the minimum and maximum of the period between 1985 and 2009 are within the range of the temperature variations between 1850 and 1984.

Base on this case study, which seems to be a rather extreme case, we expect the benthic temperatures to be roughly in agreement with observations. Also it seems plausible that the development ocean temperatures (including the benthic) on the shelves roughly follows that of the atmospheric temperatures – eventually with some delay. This effect is for the future part of our study the most important.

Also what aspects of the CLIMBER output were used to force MPI-ESM, just CO2 and other GHGs?

[Reformulation of sentence in sec. 2.6 (Boundary conditions), so that it states that only $CO_2$ is used from CLIMBER.] Only $CO_2$ was adopted from the CLIMBER simulations as boundary conditions for the MPI-ESM runs creating our benthic temperatures. $CH_4$ were calculated interactively in MPI-ESM and $N_2O$ was kept constant. See Kleinen et al. (2021) for details.

p. 3 l. 70-79. These two paragraphs seem in conflict with each other. Either salt diffusion is slow and unimportant, or it is fast and important.

[Rephrased start of 2nd paragraph in sec. 2.3 (Salinity)]Our assumption with respect to salt diffusion is that it is too slow to be important at the time scale we cover in the present study (O(1000yr)), though it is definitely play an important role for longer (i.e. glacial) time scales. We will clarify this in the revised manuscript.

p. 5, l. 136 Is this thawing under preindustrial forcing to be interpreted as lagged thaw following LGM, or is it an artifact of the imposed initial conditions?

[Sentences in the first paragraph of sec. 1 (Introduction) swapped and 2 new sentences added.]The SSPF has since it's inundation generally been overlaid by comparatively warm oceanic water, causing a steady thawing from above which – at some point in time – will cause all SSPF to be melted away (provided that the world does not enter a new glacial state causing an oceanic regression from the SSPF areas on the oceanic shelves). Therefore the thaw seen with preindustrial forcing is a lagging effect of the changed climatic conditions since the LGM. Only exception is – as discussed above – a minor additional thaw in the period ~1850-1880 which is either caused by inconsistencies between the physics of SuPerMAP and JSBACH or by

physical inconsistencies in the initial conditions of JSBACH arising from the interpolation between the vertical grids of the two models.

Since the general fate of SSPF is (independently of climate change) the basic setting of our study, we will extend the discussion and explanation in the introduction.

figs. 1, 5, &6: Is 'depth' the depth below the benthic surface or the depth below the sea surface?

["below sea floor" added to the axis of Figs. 1 and 6 (now 7) and comment in the caption of Fig. 5 (now 6).] Since we haven't dealt with the ocean model ourselves, we do not at all consider the ocean as part of our model domain. Therefore we're always having depth as "below the benthic surface/ocean bottom", which is at a fixed depth since we did not include sedimentation in the JSBACH model. We will clarify in the revised manuscript.

fig. 3: Is the ratio of melt rate to the preindustrial melt rate a meaningful metric, and if so why? It seems like the absolute loss rate is a more fundamental measure than the relative rate.  But if the relative rate is more meaningful, then some background and explanation would be helpful.

[Explanatory sentence + slight rephrasing in/of first paragraph of sec. 3 (Results).] We regard both the relative ratio (Fig. 3 in the paper and Fig. GB4) and the absolute melting (Fig. 2 in the paper and Fig. GB3) as important. The relative ratio highlights the influence of the climatic changes, since – as described above – there's always a "background" melting caused by the imbalance remaining from post-LGM climate change and inundation. This is even more highlighted by the additional scenarios pmt_pre_lc and pmt_ssp585_lc (above). Since the absolute melting via the IBPT controls the release rate of carbon ($CH_4$) to the ocean, one could phrase it as: The absolute melting states how much damage is done by the melting of SSPF and the relative ratio states how much of this damage is due to (anthropogenic) climate change.

fig. 4: It would be helpful to see some time progression here. For example how much ice will have melted by 2100, 2200, 2300, 2500, and 3000 under each of the scenarios?

[Added a comment on the pattern time independence in sec. 3 (Results).] The patterns of melting do not change much over time. Therefore we considered these interstadials as being uninteresting. However, taking into account the focus of the paper on the earlier part of the time period (say: before year 2400), it might be worth considering putting a picture of e.g. 2300 into the main text. Additional time slices we can put in the supplement.

fig. 7: Are these annual mean values, and if so, should these be thought of as the spatial area with no ice, or the fraction of the year with no ice?

[Caption refined.] These are monthly mean values, which are averages of the all the instantaneous (i.e. from each MPI-OM time step) fractional sea ice concentrations for the individual months. Thus it is a combination of an instantaneous fraction (i.e. spatial mean) and a temporal mean.

fig. 7: Why the break in slope at 70% sea ice concentration?

[Added sentence on the insulating effect above this limit to last paragraph of sec. 3 (Results).] This is hard to say in detail, since this figure basically shows only a correlation, not a causality. In MPI-OM, the sea ice concentration is assumed to be uniformly distributed over the the grid cell. This has the effect, that all sea ice has to melt away before energy is available for heating the ocean water. Our best guess is that for sea ice concentrations >70% sea ice is present for such a long time (within the month) that there's no time to heat the water. Also oceanic advection from neighbouring grid cells may play a role. Anyway: It seems obvious that there has to be a point where the insulating role of sea ice start vanishing, but we have no good explanation why it occurs at around 70%.

fig. 7: Would the plot look different if other explanatory variables (e.g. surface ocean temperature or surface air temperature) were used?

[Added the "directness" of the relation between these two variables to the last paragraph of sec. 3 (Results). Otherwise I don't see how this discussion benefits the manuscript] Since the benthic temperatures are the direct driver of the energy input into the sediments we expect a much more clear picture using this variable. The SST is – though it is very similar to the benthic temperatures except in late summer – not quite an as direct driver, and surface air temperature is even less so. A spatially limited illustration of the relation to the surface air temperature can be seen in fig. 6 of the manuscript (lower panel). This clearly shows the

decoupling of the warming of the lower atmosphere and the sediments. For a complete picture we have re-plotted Fig. 7 of the manuscript as function of SST (Fig. GB8 left) and surface air temperature (Fig. GB8 right) instead of benthic temperature. These pictures do in our minds not show any clear interesting patterns.

[Figure]

*Fig. GB8. As Fig. 7 in the manuscript, but replacing the benthic temperature with SST (left) and surface air temperature (right) respectively.*

p. 8, Line 244: does this loss also occur under a steady-state preindustrial climate, or is it a forced response to the historical warming?

[Sentences added to sec. 4.3 (Area of SSPF)] The experiment with preindustrial forcing losses about 76% of the SSPF area mentioned in the paper for the historical experiments (Fig. GB9). We attribute this initial loss mainly to loss of thin SSPF ice in the upper layers (which may in parts be an unrealistic artifact of the initialization) for which also the preindustrial forcing is sufficient to melt away. The remaining 24% is due to warming during the historical period. This interesting point will be stated in the revised paper.

[Figure]

*Fig. GB9. Pan-arctic area with SSPF and it's development over time for the scenarios presented in the manuscript. The discrete steps are due to the grid cell discretization. Total modelled area is about 4.6 mill. km$^2$ distributed on 407 points which either have or does not have SSPF.*

Fig. S1 What are the units?

[Done] The unit somehow disappeared from this figure, sorry. It is mK/m (or K/km). Will of course be corrected in the revised manuscript.

fig. S6 Is helpful in interpreting the results, I suggest moving to the main manuscript document.

[Done] This figure shows our forcing data only, which we didn't produce ourselves. We judged that we by putting it into the main manuscript risk to confuse what is actually our work and what we included from the work of others. However, since it seems to be more important than anticipated we will incorporate it in the main part of the revised manuscript.

fig. S11, the right hand side panel isn't interpretable, I think you'd need to show the amplitude for each scenario separately, and using the same color scale for both the historical and future panels.

[Done] We will redo this figure for the revised manuscript, separating the three scenarios.

Bibliography

de Vrese, P., Stacke, T., Kleinen, T., and Brovkin, V.: Diverging responses of high-latitude CO 2 and CH 4 emissions in idealized climate change scenarios, The Cryosphere, 15, 1097–1130, https://doi.org/10.5194/tc-15-1097-2021, 2021.

Ekici, A., Beer, C., Hagemann, S., Boike, J., Langer, M., and Hauck, C.: Simulating high-latitude permafrost regions by the JSBACH terrestrial ecosystem model, Geoscientific Model Development, 7, 631–647, https://doi.org/10.5194/gmd-7-631-2014, 2014.

Goto, S., Yamano, M., Morita, S., Kanamatsu, T., Hachikubo, A., Kataoka, S., Tanahashi, M., and Matsumoto, R.: Physical and thermal properties of mud-dominant sediment from the Joetsu Basin in the eastern margin of the Japan Sea, Marine Geophysical Research, 38, 393–407, https://doi.org/10.1007/s11001-017-9302-y, 2017.

Gu, X., Tenzer, R., and Gladkikh, V.: Empirical models of the ocean-sediment and marine sediment-bedrock density contrasts, Geosciences Journal, 18, 439–447, https://doi.org/10.1007/s12303-014-0015-9, 2014.

Jungclaus, J. H., Fischer, N., Haak, H., Lohmann, K., Marotzke, J., Matei, D., Mikolajewicz, U., Notz, D., and von Storch, J. S.: Characteristics of the ocean simulations in the Max Planck Institute Ocean Model (MPIOM) the ocean component of the MPI-Earth system model, Journal of Advances in Modeling Earth Systems, 5, 422–446, https://doi.org/https://doi.org/10.1002/jame.20023, 2013.

Jungclaus, J. H., Keenlyside, N., Botzet, M., Haak, H., Luo, J. J., Latif, M., Marotzke, J., Mikolajewicz, U., and Roeckner, E.: Ocean circulation and tropical variability in the coupled model ECHAM5/MPI-OM, JOURNAL OF CLIMATE, 19, 3952–3972, https://doi.org/10.1175/JCLI3827.1, 2006.

Kleinen, T., Gromov, S., Steil, B., and Brovkin, V.: Atmospheric methane underestimated in future climate projections, ENVIRONMENTAL RESEARCH LETTERS, 16, https://doi.org/10.1088/1748-9326/ac1814, 2021.

Mauritsen, T., Bader, J., Becker, T., Behrens, J., Bittner, M., Brokopf, R., Brovkin, V., Claussen, M., Crueger, T., Esch, M., Fast, I., Fiedler, S., Flaeschner, D., Gayler, V., Giorgetta, M., Goll, D. S., Haak, H., Hagemann, S., Hedemann, C., Hohenegger, C., Ilyina, T., Jahns, T., Jimenez-de-la Cuesta, D., Jungclaus, J., Kleinen, T., Kloster, S., Kracher, D., Kinne, S., Kleberg, D., Lasslop, G., Kornblueh, L., Marotzke, J., Matei, D., Meraner, K., Mikolajewicz, U., Modali, K., Moebis, B., Muellner, W. A., Nabel, J. E. M. S., Nam, C. C. W., Notz, D., Nyawira, S.-S., Paulsen, H., Peters, K., Pincus, R., Pohlmann, H., Pongratz, J., Popp, M., Raddatz, T. J., Rast, S., Redler, R., Reick, C. H., Rohrschneider, T., Schemann, V., Schmidt, H., Schnur, R., Schulzweida, U., Six, K. D., Stein, L., Stemmler, I., Stevens, B., von Storch, J.-S., Tian, F., Voigt, A., Vrese, P., Wieners, K.-H., Wilkenskjeld, S., Winkler, A., and Roeckner, E.: Developments in the MPI-M Earth System Model version 1.2 (MPI-ESM1.2) and Its Response to Increasing CO2, JOURNAL OF ADVANCES IN MODELING EARTH SYSTEMS, 11, 998–1038, https://doi.org/10.1029/2018MS001400, 2019.

---

## Author Response (AR2)

Hamburg 14/2 2022

Dear editor and dear reviewers,

We, the authors are very happy that you are satisfied by our revisions and answers to your comments. Also we are very grateful that the editor found our work so interesting that it has been selected as a highlight contribution to The Cryosphere.

A single editorial comment: For both uploaded versions of our manuscript, I recieved the (more or less automatically generated) message "With the next revision, please remove "Wilkenskjeld Miesner Overduin Puglini Brovkin" from page 1.". This list of author names is inserted by your own LaTeX template, and thus I would not know how to remove it. I hope this is not a problem for the final typesetting.

We thank for the corporation.

Best regards

Stiig Wilkenskjeld on behalf on all authors.